# CHARACTERIZING RESNET'S UNIVERSAL APPROXIMATION CAPABILITY

## ABSTRACT

Since its debut in 2016, ResNet has become arguably the most favorable architecture in deep neural network (DNN) design. It effectively addresses the gradient vanishing/exploding issue in DNN training, allowing engineers to fully unleash DNN's potential in tackling challenging problems in various domains. Despite its practical success, an essential theoretical question remains largely open: how well can ResNet approximate functions? In this paper, we provide answers to this question. We first show that ResNet with bottleneck blocks (b-ResNet) can approximate any $d$-dimensional monomial with degree $p \geq d$ to any accuracy $\varepsilon > 0$ with $\mathcal{O}(p \log(p/\varepsilon))$ number of tunable weights. We extend the results to show that smooth functions differentiable up to degree $r$ can be approximated by ResNet using $\mathcal{O}(c(d,r)\varepsilon^{-d/r} \log 1/\varepsilon)$ tunable weights, where $c(d,r)$ is a constant depending on $d$ and $r$. This is a factor of $d$ reduction in the number of tunable weights compared with the classical results for ReLU networks. Furthermore, our results are order-optimal in $\varepsilon$ as they match a generalized lower bound derived in this paper. Finally, we show that ResNet can approximate a class of smooth functions, by leveraging the Kolmogorov Superposition Theorem, using $\mathcal{O}(d^4\varepsilon^{-1})$ tunable weights, which is only a polynomial (instead of exponential) function of $d$, implying no curse of dimensionality for approximating this class of functions.

## 1 INTRODUCTION

One trend in deep learning in the past decade is the use of larger and deeper neural networks to process higher-dimensional data. However, for very deep neural networks, training can lead to serious optimization problems, such as gradient vanishing or exploding. In 2016, the emergence of ResNet He et al. (2016) solved the problem of gradient vanishing or exploding encountered during the training of deep neural networks and has shown outstanding performance in applications.

The practical success of ResNet naturally leads to an essential theoretical question: how well can ResNet approximate functions? Along this line, a milestone result in Lin & Jegelka (2018) shows universal approximation capability of ResNet (even with one neuron per layer): it can approximate any Lebesgue-integrable function arbitrarily well as the number of tunable weights goes to infinity. This result gives theoretical justification to using ResNet to approximate general functions and spurs a number of follow-up studies. One of the most related works is Oono & Suzuki (2019) which focuses on the approximation of ResNet-type convolutional neural networks (CNNs) but the results can not apply to ResNet-type multilayer perceptrons(MLPs) in Lin & Jegelka (2018). See Sec. 1.1 for more discussions about the key differences in settings, results and methods with our work. In this paper, unless otherwise specified, we will refer to ResNet as ResNet-type MLPs. (See Sec. 2 for specific math modeling.) Meanwhile, it remains largely open to characterize ResNet's universal approximation capability. That is, *how many tunable weights are needed for ResNet with optimized structures to approximate a function up to an error $\varepsilon$?*

In this paper, we seek answers to the above question, by developing upper-/lower- bounds on tunable weights for ResNet with bottleneck block (b-ResNet for short) to approximate several popular classes of functions. We summarize the **main contributions** as follows:

▷ In Sec. 3, we explicitly establish the relationship between ResNet and feedforward networks (FNNs) (see Proposition 1). We show that ResNet can be viewed as an FNN and thus derive lower bounds on the number of tunable parameters for ResNet to approximate various classes of functions.

$\triangleright$ In Sec. 4, we show that b-ResNet, by leveraging specific tunable weights, can approximate various function classes. These include monomials with degree $p$, requiring the number of weights $\mathcal{O}(p \log p/\varepsilon)$, polynomials of degree $p$, needing $\mathcal{O}(p\#\text{terms} \log p/\varepsilon)$[1], and smooth functions differentiable up to degree $r$, needing $\mathcal{O}_{d,r}(\varepsilon^{-d/r} \log 1/\varepsilon)$[2]. In addition, we show ResNet with one neuron per hidden layer can generate any continuous piece-wise linear function thereby approximating continuous functions.

$\triangleright$ In Sec. 5, we apply ResNet to approximate a distinctive class of functions derived from Kolmogorov Superposition Theorem (KST), using $O(d^4\varepsilon)$ tunable weights. This approach circumvents the so-called curse of dimensionality. Subsequently, in Sec. 6, we give the simulation results that validate the theoretical findings in this paper.

These findings add to the theoretical justifications for ResNet's outstanding practical performance, and shed light on neural network architecture and analysis for further research on NN design optimization.

## 1.1 RELATED WORK

In recent years, the expressive capabilities of various neural network architectures have garnered increased attention, spurred by their remarkable and noteworthy successes. In this subsection, we will discuss the previous research on various network architectures through the lens of approximation theory. **Due to space limitations, more detailed information on the related work can be found in Appendix G.1.**

**Universality and Approximation Capabilities.** The universality of neural networks is a topic of considerable importance and substantial interest. Its exploration spans from shallow Cybenko (1989); Pinkus (1999) to deep Hanin & Sellke (2017); Kidger & Lyons (2020) networks, and from MLPs to other network structures including standard deep ReLU CNN Zhou (2018; 2020), deep ReLU CNNs with classical structures He et al. (2022), continuous-time recurrent neural network (RNN) Li et al. (2020; 2022b), continuous-time ResNet Li et al. (2022a), ResNet Lin & Jegelka (2018), and ResNet for finite-sample classification tasks Hardt & Ma (2016). Over the past decades, there has been substantial progress in enhancing our theoretical understanding of neural networks from various views: the benefit of depth Arora et al. (2016); Eldan & Shamir (2016); Liang & Srikant (2016); Telgarsky (2016); Yarotsky (2017); Poggio et al. (2017), combinatorics Montufar et al. (2014); Serra et al. (2018); Arora et al. (2016), approximation capabilities Shen et al. (2022b); Yarotsky (2018; 2017); Lu et al. (2021); Montanelli & Du (2019); Wang et al. (2018); Schwab & Zech (2021).

**Perspectives on the Curse of Dimensionality.** The 'curse of dimensionality' coined by Bellman (1957) refers to a phenomenon that a model class will suffer an exponential increase in its complexity as the input dimension increases. This impact is explicitly observed in ReLU networks, as well-documented in Yarotsky (2017). Importantly, the curse of dimensionality, not limited to MLPs, is also a challenge for almost all classes of function approximators aiming to uniformly approximate in the Lipschitz domain (have sufficient regular boundary) on some compact subset of a metric space due to the entropy limitation Kolmogorov & Tikhomirov (1959).

More specifically, any continuous function approximator[3] will suffer the curse of dimension in the smooth function space $C^r$ DeVore et al. (1989) because the metric entropy of the unit ball in $C^r$ with respect to the uniform topology is $\Theta(\varepsilon^{-d/r})$. The property is applied to ReLU neural networks in Thm. 3 Yarotsky (2017). In an attempt to mitigate the curse of dimensionality, initial strategies involved the consideration of specialized function spaces whose metric entropy is expected to reduce such as analytical functions Wang et al. (2018), bandlimited functions Montanelli et al. (2019), Korobove space Montanelli & Du (2019).

---

[1] The notation '#' is an abbreviation of number and '#terms' refers to the number of terms in the polynomial.

[2] The notation $a = \mathcal{O}(g(\varepsilon))$ means $a \leq Cg(\varepsilon)$ for sufficiently small $\varepsilon$ where $C$ is a constant independent of $\varepsilon$. Importantly, throughout this paper, we employ the notation $\mathcal{O}_d(\cdot)$ to underscore the hidden constant $C$ depending on $d$.

[3] In the context, we aim to approximate all functions in a space $\mathcal{F}$ using a model class as an approximator (e.g., neural networks). We achieve this by choosing different parameters for different functions, meaning the parameters $\theta \in \Theta$ can be seen as a mapping of the target functions, i.e., $\theta = h(f)$ where $h : \mathcal{F} \to \Theta$. If this mapping $h$ is continuous, we refer to the approximator as a continuous approximator.

More recently, researchers have shifted their focus toward the structure of neural networks, suggesting a potential solution to circumvent the curse of dimensionality. One approach involves the parameters-sharing method (e.g., repeated-composition structure Zhang et al. (2023), CNN Zhou (2018)). Simultaneously, a more recent trend aims to serve neural networks as discontinuous function approximators, thereby examining neural networks with novel activation functions (e.g. Shen et al. (2020); Jiao et al. (2023); Shen et al. (2021; 2022a)). Meanwhile, some incorporate the Kolmogorov Superposition Theorem (KST)(see Sec. 5) to tackle the curse of dimensionality, yielding promising theoretical results: considering super-expressive activations Yarotsky (2021); Shen et al. (2022a) or restricted function class derived from KST Lai & Shen (2021); He (2023) can be efficient strategies to avoid the curse of dimensionality. However, the failure of these model classes in practice is due to the discontinuity of the function approximators, wherein even minor perturbations in the training data can lead to chaotic changes in the input-output relationship. Consequently, to circumvent the curse of dimensionality, it is imperative to make appropriate choices within the unstable model class and the restricted objective function space.

**Relations with ResNet-type CNNs.** Following the work of Lin & Jegelka (2018), both Oono & Suzuki (2019) and our study focus on the approximation error of ResNet-type structure neural networks. However, several key differences in the settings, results, and methodologies are worth noting.

First, the settings and neural network architectures are different. Our work zeroes in on FNNs in ResNet which aligns close to Lin & Jegelka (2018), while Oono & Suzuki (2019) investigates CNNs in ResNet which is more closely aligned with Zhou (2018); Petersen & Voigtlaender (2020). FNN and CNN possess distinct network structures, and our analysis, just like that of Oono & Suzuki (2019), leverages the unique structures of FNN and CNN, respectively. Consequently, results derived from one do not directly translate to the other.

Second, the approaches we adopted diverge significantly. Oono & Suzuki (2019) illustrates that any block-sparse FNN can be realized by ResNet-type CNN, and they leverage this relationship to set the upper bounds of the approximation capability of ResNet CNNs. In contrast, we demonstrate that ResNet FNN can be implemented by ReLU FNN Prop. 1, and we utilize this relationship to set the lower bounds on the approximation capability of ResNet FNN. We establish the upper bounds by directly constructing ResNet FNN to approximate different function classes.

Third, the results are different. Our paper offers a novel construction of a simplified ResNet FNN structure, known as b-ResNet, and comprehensively delineates both the lower and upper bounds of its function approximation capability. Notably, we demonstrate that it achieves comparable approximation accuracy to ReLU FNN but with a significantly reduced number of tunable weights (a reduction by a factor of $d$, where $d$ is the input dimension). In contrast, Oono & Suzuki (2019) establishes that a ResNet CNN can attain the same approximation accuracy as block-sparse FNNs, albeit with an order-wise equivalent number of tunable weights. However, it remains uncertain from Oono & Suzuki (2019) whether a ResNet CNN would require fewer weights.

## 2 MATHEMATICAL MODELING OF RESNET AND PROOF IDEAS

### 2.1 MATHEMATICAL MODELING OF RESNET

ResNet's original proposal He et al. (2016) includes complex structures such as convolutional layers. In this paper, we focus on its basic structure which consists of residual blocks and identity shortcut connections, and we show that it is enough to achieve a strong approximation capability. With the basic notations of addition $(f + g)(x) = f(x) + g(x)$ and composition $f \circ g(x) = f(g(x))$ of mappings $f, g$, a ResNet is a function $R(x)$ from $\mathbb{R}^d$ to $\mathbb{R}$ given by

$$R(x) = \mathcal{L} \circ (\mathcal{T}^{[L]} + Id) \circ \cdots \circ (\mathcal{T}^{[1]} + Id) \circ \mathcal{A}_k(x), \tag{1}$$

where $\mathcal{A}_k : \mathbb{R}^d \to \mathbb{R}^k$ and $\mathcal{L} : \mathbb{R}^k \to \mathbb{R}$ are affine transformations, $Id : z \mapsto z$ is the identity mapping, and $\mathcal{T}^{[i]}(i = 0, 1, ..., L)$ are basic residual blocks. Each block $\mathcal{T}^{[i]} : \mathbb{R}^k \to \mathbb{R}^k$ further consists of two layers:

- an activation layer $\mathbb{R}^k \to \mathbb{R}^{n_i} : z \mapsto \sigma(W_i z + b_i)$ with $n_i$ neurons, and
- an identity layer $\mathbb{R}^{n_i} \to \mathbb{R}^k : \sigma(W_i z + b_i) \mapsto V_i \sigma(W_i z + b_i)$ with $k$ neurons,

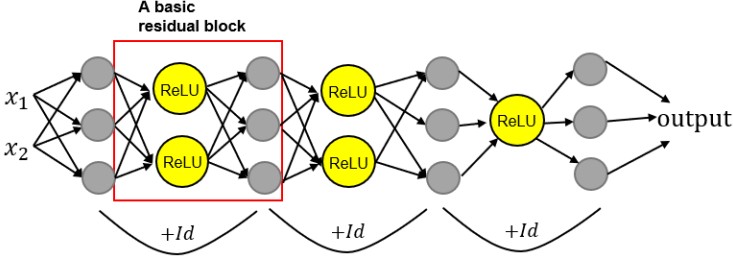

Figure 1: An example of a ResNet from $\mathbb{R}^2$ to $\mathbb{R}$ belonging to $\mathcal{RN}\,(k=3,N=2,L=3)$. Every residual block is composed of an activation layer followed by an identity layer. The activation layer neurons are colored yellow, while the identity layer neurons are grey. The maximum number of neurons in each activation layer is 2, each identity layer has 3 neurons, and it has 3 blocks.

where parameters $W_i \in \mathbb{R}^{n_i \times k}, V_i \in \mathbb{R}^{k \times n_i}, b \in \mathbb{R}^{n_i}$ $(i = 1, 2, \cdots, L)$, and $\sigma(\cdot)$ is the generalized ReLU activation function for vector output, i.e., $\sigma(x_1, x_2, \cdots, x_n) = (\max 0, x_1, \cdots, \max 0, x_n)$ for $x_1, x_2, \cdots, x_n \in \mathbb{R}$. To this end, we write $\mathcal{T}^{[i]}(z) = V_i\sigma(W_i z + b_i)$.

We use $L$ to denote ResNet's depth defined as the number of residual blocks, and its width $N$ is defined as the maximum number of **activation layer neurons**, i.e., $\max\{n_1, n_2, \cdots, n_L\}$. For conciseness of notation, we denote by $\mathcal{RN}\,(k, N, L)$ the set of ResNet functions from $\mathbb{R}^d$ to $\mathbb{R}$, with width $N$, depth $L$, and $k$ neurons in each identity layer. To keep our study interesting, we always assume that $k \geq d$ where $d$ is the input dimension. Otherwise, the universality of the ResNet does not hold. [4]. For our later discussion, we define a ResNet as **ResNet with bottleneck blocks (b-ResNet)** if it belongs to $\mathcal{RN}\,(k, N = C, L)$ where $C$ is an absolute constant (much smaller than and independent of input dimension $d$). We have an example of the ResNet structure defined above in Fig. 1.

## 2.2 PROBLEM STATEMENT

Lin & Jegelka (2018) shows the universal approximation capability for ResNet with even one neuron per activation layer in each block: it can approximate any Lebesgue-integrable function with arbitrary error as the depth $L$ goes to infinity. Thus, we need to characterize ResNet's universal approximation capability to solve the open question: how many tunable weights are needed for general ResNet to approximate a function to an error. We give the following problem statement.

Given $f^\star : [0, 1]^d \to \mathbb{R}$ belong to some function space $\mathcal{F}$ and fix $d \in \mathbb{N}_+$ and the ResNet model $\mathcal{H} = \mathcal{RN}\,(k, N, L)$, we are interested in the following two questions specifically.

▷ Lower bound. What is the minimum number of weights of ResNet required to approximate any $f^\star \in \mathcal{F}$ to an error $\varepsilon$?

▷ Upper bound. What is the number of weights of a ResNet architecture sufficient to approximate any $f^\star \in \mathcal{F}$ to an error $\varepsilon$? Or does there exists $k, N, L$ such that $\inf_{\tilde{f} \in \mathcal{H}} \left\| \tilde{f} - f^\star \right\|_\infty \leq \varepsilon$ holds with relatively smaller number of tunable parameters?

Here tunable parameters refer to non-zero parameters in ResNet. Note the answer to the two questions usually depends on the desired error $\varepsilon$, the input dimension $d$, and the function space $\mathcal{F}$. In this paper, we use uniform norms ($\| \cdot \|_\infty$) to measure the distance between functions. Our main results show that $\mathcal{RN}\,(k, N, L)$ with $k = d + c_1, N = c_2$ ($c_1, c_2$ are absolute constants) and proper $L$ can achieve powerful approximation capability in various function classes. It can approximate specific functions with fewer tunable functions than that of FNNs.

---

[4]If $k < d$, the ResNet with one neuron per activation layer belongs to the set of narrow networks with widths smaller than or equal to $d$ (Prop. 1). Therefore, the ResNet may lose the universality since narrow-width (smaller than $d + 1$) ReLU networks cannot approximate all $\mathbb{R}^d$ continuous functions Hanin & Sellke (2017).

### 2.3 PROOF IDEAS AND NOVELTY

We outline the proof ideas as follows.

**Lower Bound.** We establish that ResNet can be conceptualized as a sparse FNN (Prop. 1), implying that the lower bound of ResNet must exceed that of FNN. We then derive the generalized lower bounds of ResNet on the approximation of various function class from the lower bounds of FNNs.

**Upper Bound.** Drawing inspiration from the work of Yarotsky (2017), where DNNs are constructed to approximate the function $x^2$ and $xy$, we construct ResNet to approximate these fundamental functions. Then any polynomial can be expressed as a composition of the product function $xy$, and smooth functions can be approximated by polynomials due to the local Taylor expansion property. By constructing ResNet to approximate these functions, we can derive upper bounds on the approximation of polynomials and smooth functions. Our results show that b-ResNet is enough to approximate these smooth functions with fewer tunable weights than that of FNN.

Our novelty lies in the theoretical exploration of the approximation capabilities of ResNet, specifically b-ResNet. Our proof utilizes a new construction of ResNet blocks for approximating primitive functions $x^2$ and $x \cdot y$. Furthermore, the role of identity mappings is greatly leveraged in the construction. Moreover, we extend the result that ResNet with one neuron per activation layer can generate any step functions, to piecewise-linear function through constructive methods, then providing an upper bound approximation for continuous functions. We uncover the extensive expressive power of b-ResNet, providing theoretical guarantees and insights into the successful performance of ResNet.

## 3 LOWER BOUNDS ON RESNET'S FUNCTION APPROXIMATION CAPABILITY

In this section, we build the explicit relation between ResNet and FNNs to establish the lower bound on the complexity of ResNet to approximate polynomials and other function classes including smooth functions and continuous functions space. The lower bounds help us to analytically show our upper bounds are optimal in terms of $\varepsilon$ in Sec. 4.

First, we propose a key argument that a ResNet can be regarded as a special sparse ReLU network.

**Proposition 1.** *For any ResNet $R(x) \in \mathcal{RN}(k, N, L)$ of the input $x \in [0, 1]^d$ with $W$ number of tunable parameters, there exists an equivalent ReLU FNN $\Phi(x) : [0, 1]^d \to \mathbb{R}$ with width $N + k$, depth $2L$, $\Theta(W + kL)$ number of tunable parameters, such that $R(x) = \Phi(x)$, $\forall x \in [0, 1]^d$.*

This proposition implies that if a ResNet can approximate a function up to an error $\varepsilon$, then an FNN with a larger size can also approximate the same function up to $\varepsilon$. Thus one can bound ResNet's universal approximation capability by studying that of a larger-size FNN (might have an increase in tunable parameters by a factor of $d$). That said, the lower bound of the complexity of ResNet must be larger than or equal to that of FNN **in terms of** $\varepsilon$ when approximating the same function.

Thus, building on the above discussion, lower bounds for FNNs in the existing literature can be applied to ResNets in terms of $\varepsilon$. We discuss in the following. Regarding the smooth space $C^r([0, 1]^d)$, Yarotsky (2017) established lower bounds of $\Theta_r(\epsilon^{-d/r})^5$ for continuous ReLU network approximators and $\Theta_{r,d}(\epsilon^{-d/2r})$ for unconstrained deep ReLU networks. Later, Yarotsky (2018) demonstrated optimal error approximation rates of $\mathcal{O}_d(\omega_f(W^{-1/d}))$ for continuous ReLU network approximators, and $\mathcal{O}_d(\omega_f(W^{-2/d}))$ for unconstrained deep ReLU networks, where $W$ represents the number of tunable weights and $\omega_f(t) := \sup |f(x) - f(y)| : |x - y| \leq t$ is the modulus of continuity. If $f$ belongs to the Lipschitz continuous function class, the optimal upper bound for the ReLU FNN is $\mathcal{O}_d(\varepsilon^{-d})$ for continuous weight selection, and $\mathcal{O}_d(\varepsilon^{-d/2})$ for unconstrained deep networks. We will return to these discussions in the next section. Yet, for the polynomial function space (a smaller space compared to continuous and smooth functions), the aforementioned lower bound is not applicable and the lower bound on the approximation of polynomials is not available in the existing literature although the proof is simple. Below we give the lower bound of the complexity of ResNet on the approximation of polynomials where the purpose is to show the upper bound in Thm. 4 is $\varepsilon$-order optimal in the next section.

---

[5]Here $\Theta_d(g(\varepsilon)) \sim Cg(\varepsilon)$ for sufficient small $\varepsilon$ where $C$ is some constant which can not depend on $\varepsilon$ but depend on $d$. The subscript $d$ on $\Theta$ emphasizes that the constant $C$ may depend on $d$.

We begin with some notations. Let $x \in \mathbb{R}^d$ and $\alpha = (\alpha_1, \alpha_2, \cdots, \alpha_d)$ where $\alpha_i \in \mathbb{N}$. Define $x^\alpha = x_1^{\alpha_1} x_2^{\alpha_2} \cdots x_d^{\alpha_d}$ and this is called a monomial. The degree of the monomial is $|\alpha| := \alpha_1 + \alpha_2 + \cdots + \alpha_d$. Then a multivariate polynomial is a sum of several monomials and its degree is the highest degree among these monomials. Then we give the theorem.

**Theorem 2.** *Let $x = [x_1, x_2, \ldots, x_d] \in [0,1]^d$ and $\mathcal{P}(d,p)(p \geq d)$ be the set of $d$ dimension polynomial functions with degree $p$. If a ReLU FNN $\Psi(x) : [0,1]^d \to \mathbb{R}^d$ with width $N$, depth $L$ and $T = NL$ neurons can approximate any $f \in \mathcal{P}(d,p)$ to an error $\varepsilon$, i.e.*

$$|\Psi(x) - f(x)| < \varepsilon, \forall x \in [0,1]^d,$$

*then we have $T \geq \Theta_d(\log 1/\varepsilon)$. Note this lower bound can also be applied to ResNet.*

## 4 UPPER BOUNDS ON RESNET'S FUNCTION APPROXIMATION CAPABILITY

In this section, we present theoretical results on the approximation capability of ResNet. Our main result is that b-ResNet can approximate polynomials and smooth functions with fewer tunable parameters than those of FNNs. Subsection 4.1 presents the upper bounds for ResNet on approximating monomials. Subsequently in subsection 4.2 we extend the results to polynomials and smooth functions in Sobolev space following the work of Yarotsky (2017). In subsection 4.3, the properties of ResNet on approximating CPwL functions are discussed, followed by the upper bound of ResNet on the approximation of continuous functions.

### 4.1 APPROXIMATING MONOMIALS

Our first key result shows that b-ResNet $\mathcal{RN}(k = \mathcal{O}(d), N = C, L)$ can approximate any monomials where $L$ depends on $d$ and the desired epsilon $\varepsilon$, and $C$ is an absolute constant independent of $d$. Moreover, this result explains why b-ResNet can approximate polynomials and smooth functions with fewer tunable parameters.

**Theorem 3.** *Let $x = [x_1, x_2, \cdots, x_d] \in [-M, M]^d$, $M \geq 1$, $\alpha \in \mathbb{N}^d$ and $x^\alpha$ be any given monomial with degree $p$, i.e., $|\alpha| = p$. Then there exists a b-ResNet*

$$R \in \mathcal{RN}\left(d + 3, 4, \mathcal{O}(p \log(p/\varepsilon) + p^2 \log M)\right)$$

*such that*

$$\|R - x^\alpha\|_{C([-M,M]^d)} < \varepsilon$$

*while having $\mathcal{O}\left(p \log(p/\varepsilon) + p^2 \log M\right)$ tunable weights.*

Note that the number of total weights of $R$ is $\mathcal{O}(dp \log(p/\varepsilon))$ when $M = 1$. However, in each constructive residual block, there are only absolute constant non-zero weights as shown in our constructive proof. Thus it is enough to adjust $\mathcal{O}(p \log(p/\varepsilon))$ weights. For $M > 1$, we have the same analysis. Moreover, it is easy to see that the upper bound in the above theorem is independent of $d$. Actually, the dimension $d$ is embedded to $p$ because any monomial with degree $p$ can always be viewed as a product function of dimension $p$. For example, $x_1^2 x_2 x_3^2 = \pi(x_1, x_1, x_2, x_3, x_3)$ where $\pi(x_1, \cdots, x_d) = x_1 x_2 \cdots x_d$ is the product function.

We highlight several key observations and discussions from the analysis and results in the following.

**ResNet vs FNNs.** We show that ResNet is capable of approximating any monomial with degree $p$ on $[0,1]^d$ with $\mathcal{O}\left(p \log(p/\varepsilon)\right)$ number of tunable weights, surpassing the upper bound of ReLU FNNs. According to DeVore et al. (2021), a ReLU network with width $\mathcal{O}(d)$ and depth $\mathcal{O}\left(p \log(p/\varepsilon)\right)$ can approximate any monomial with degree $p$, resulting in a total weight count of $\mathcal{O}\left(d^2 p \log(p/\varepsilon)\right)$. According to their construction, there are $\mathcal{O}\left(dp \log(p/\varepsilon)\right)$ tunable weights (non-zero weights). Thus, our result has a reduction by a factor $d$. Notably, our upper bounds on the approximate value of product functions and monomials are optimal to $\varepsilon$ according to the lower bound in Theorem 2.

**Root of reduction**. Note that each identity mapping can be realized by $2d$ ReLU units ($x = [x]_+ - [-x]_+$) but it only may lead to an 'additive' reduction of $d$ tunable weights compared to an FNN. In our study, however, this additive reduction of $d$ tunable weights, as seen in our b-ResNet model, does translate into a multiplicative reduction by a factor of $d$. We establish this result by

proving that a b-ResNet with a constant number of tunable weights per residual block can approximate functions with the same accuracy as a ReLU FNN requiring $O(d)$ tunable weights. For the reason why identity mappings between layers play a big role in b-ResNet, we conduct a brief analysis. Assume that the input of a block is a $d$-dimensional vector consisting of linearly independent elements. The initial layer (activation layer) of a bottleneck block reduces the dimension to a smaller number $N \ll d$, and the following layer (identity layer) reinstates the width to $d$. The absence of identity mapping will make the value of each neuron in the identity layer be the linear combination of $N$ neurons in the previous layer, thereby significantly reducing the effective width of this layer to $N$. Nevertheless, the inclusion of identity mapping enables the recovery of linear independence for the $d$ neurons in this layer, thus endowing them with exceptional expressive capabilities even with fewer weights. More constructive details can be found in Appendix C.

**Deep vs Shallow.** Shapira (2023) provide a lower bound on the complexity of shallow FNNs to approximate any non-normalized monomial over $[-M, M]^d$ which scales exponentially with $d$ (refer to Thm. 3 Shapira (2023)). By proposition 1, this lower bound also applies to shallow ResNet. Conversely, Theorem 3 gives a mild upper bound for deep ResNet which scales polynomially with $d$. This underscores the benefits of deep networks. The problem of depth-width tradeoffs in FNNs has been discussed extensively in classical literature (e.g., Safran & Shamir (2017); Liang & Srikant (2016)). However, in this paper, we have not delved into the depth-width tradeoffs for ResNets due to our focus on constructing b-ResNets (with constant width). A more extensive exploration is a valuable and interesting future direction.

### 4.2 APPROXIMATING POLYNOMIALS AND SMOOTH FUNCTIONS

Polynomials are the summation of monomials and a smooth function can be approximated by a polynomial as per the local Taylor expansion. In this subsection, we display the upper bounds on the approximation of polynomials (Thm. 4) and smooth functions (Thm. 5).

**Theorem 4.** *Let $x = [x_1, x_2, \cdots, x_d] \in [0, 1]^d$. For a multivariate polynomial $P(x)$ with degree $p$, i.e., $P(x) = \sum_{\alpha \in E} c_\alpha x^\alpha$ where $E = \{\alpha \in \mathbb{N}^d : |\alpha| \leq p\}$, there exists a ResNet*

$$R \in \mathcal{RN}\left(d + 4, 4, \mathcal{O}\left(p|E| \log\left(p/\varepsilon\right)\right)\right)$$

*such that*

$$|R(x) - P(x)| < \varepsilon \cdot \sum_{|\alpha| \in E} |c_\alpha|, \quad \forall x \in [0, 1]^d.$$

*Additionally, the ResNet has $\mathcal{O}(p|E| \log(p/\varepsilon))$ tunable weights.*

Note $|E| = \binom{p+d}{p}$ which exponentially increase in $d$ when $p$ or $d$ is very large. Nonetheless, we can see this upper bound is optimal in terms of $\varepsilon$ according to Thm. 2.

The Sobolev space $W^{r,\infty}([0,1]^d)$ is the set of functions belonging to $C^{r-1}([0,1]^d)$ whose $(r-1)$-th order derivatives are Lipschitz continuous. Further definitions can be found in Appendix D. We then give the upper bounds of ResNet's complexity in the following theorem.

**Theorem 5.** *Fix $r, d \in \mathbb{N}_+$. There is a ResNet $R(x) \in \mathcal{RN}\left(d + 4, N = 4, L\right)$ that can approximate any function from the unit ball of $W^{r,\infty}([0,1]^d)$ to $\varepsilon$ where $L = \mathcal{O}_{d,r}(\varepsilon^{-d/r} \log 1/\varepsilon)$.*

The number of tunable parameters is still less than that of FNN by a factor $d$ based on the polynomial approximation methods. However, the hidden constant $c(d, r)$ in $\mathcal{O}_{d,r}(\varepsilon^{-d/r} \log 1/\varepsilon)$ is very large in $d$ where an estimation is given as $(\frac{2^{\frac{d+1}{r}}d}{r})^d < c(d, r) < (\frac{2^{\frac{d+1}{r}}d}{r})^d d^{r+2}(d+r)r$. Moreover, Yarotsky (2017) established the lower bound $\Theta_r(\epsilon^{-d/r})$ of the tunable weights for continuous ReLU network approximators on the approximation of the Sobolev space $W^{r,\infty}([0,1]^d)$. Based on Prop. 1, the lower bound can also apply to ResNet. Thus, we can see the upper bound in Thm. 5 is nearly tight up to a $\log$ factor for ResNet. As a closing remark, we acknowledge that while the Sobolev space is critical, it is a limited function class.

### 4.3 APPROXIMATING CONTINUOUS FUNCTIONS

In this subsection, we show that ResNet, a type of narrow network, can generate any continuous piecewise linear (CPwL) functions, and furthermore, we establish the upper bound on ResNet's complexity to approximate continuous functions over $[0, 1]^d$.

**Theorem 6.** *For any CPwL function $f : \mathbb{R}^d \to \mathbb{R}$, there exists a ResNet $R(x) \in \mathcal{RN}(d+1, 1, L)$ with $L = \mathcal{O}(Md)$ that can exactly represent $f$ where $M$ is an $f$-dependent number.*

*Moreover, for any given continuous function $f : [0,1]^d \to \mathbb{R}$, one can use ResNet with one neuron per activation layer to approximate $f$ within $\varepsilon$ with $\mathcal{O}_d(\omega_f(\varepsilon)^{-d})$ tunable weights by piece-wise spline approximation where*

$$\omega_f(t) := \sup\{|f(x) - f(y)| : |x - y| \le t\}.$$

Note that $M$ is implict which is an $f$-dependent number. It depends on the property of the input CPwL function $f$ including the number of pieces and linear components. More details about the representation of CPwL functions can be found in Tarela & Martinez (1999); Wang & Sun (2005) Moreover, a recent work Chen et al. (2022) derived a dimension-independent bound for ReLU networks if the number of pieces and linear components of the target CPwL function is known. We conjecture that ResNet can also satisfy the property by similar methods and leave it to future work. Piecewise linear spline approximation holds a significant position in approximation theory, as it is a basic method of approximating functions. Therefore, studying the expressive power of neural networks for piecewise linear functions becomes particularly important. More discussions can be found in Appendix E.2.

Theorem 6 provides another approach to demonstrating the universal theorem. While Lin & Jegelka (2018) shows that ResNet with one neuron per activation layer can approximate any step function, we extend step functions to any CPwL function. When we use a one neuron per activation layer ResNet to approximate continuous functions over $[0,1]^d$, the weights are at most $\mathcal{O}_d(\omega_f(\varepsilon)^{-d})$. Based on the discussion in Sec. 3, it is worth noting that the upper bound $\mathcal{O}_d\left(\omega_f(\varepsilon)^{-d}\right)$ is tight to $\varepsilon$ among continuous approximators.

Finally, we briefly comment on the limitations of our results. In Sec. 3, we have discussed that the optimal approximation rates in Sobolev space $W^{r,\infty}([0,1]^d)$ and continuous function space $C([0,1]^d)$ achieved by unconstrained deep ReLU FNNs, should be $\mathcal{O}(W^{-2r/d})$ and $\mathcal{O}(\omega_f(W^{-2/d}))$ respectively Yarotsky (2018); Yarotsky & Zhevnerchuk (2020); Shen et al. (2022b). Our results are suboptimal by an exponential factor of $1/2$ due to polynomial approximation methods compared with these more modern quantization-based arguments. Even though b-ResNet can achieve the suboptimal approximation rate, it remains uncertain whether it can reach the optimal approximation rate. However, drawing from similar methods to Shen et al. (2022b) which are possible for ResNet structure, we conjecture this non-trivial and important extension is feasible and we leave them in future work.

## 5 APPLICATION ON KST STRUCTURE

Building on previous conclusions, we identify the curse of dimensionality in approximating polynomials, smooth functions, and continuous functions. In this section, we present Kolmogorov Superposition Theorem and illustrate how to leverage ResNet to avoid the curse of dimensionality within a specific function class dense in $C([0,1]^d)$.

**Theorem 7** (Kolmogorov Superposition Theorem (KST),1957)**.** *Every continuous function $f$ defined on $[0,1]^d$ can be represented in the form:*

$$f(x_1, \cdots, x_d) = \sum_{q=0}^{2d} g\left(\sum_{i=1}^{d} \lambda_i \phi_q(x_i)\right) \tag{2}$$

*where $\lambda_p$ are constants for $p = 1, \cdots, d$ satifying $0 < \lambda_p, \sum_{p=1}^{d} \lambda_p \le 1$, $\phi_q(x)$ are strictly increasing Lipschitz functions (Lip < 1) (independent of $f$ ) defined on $[0,1]$ for $q = 0, \cdots, 2d$ and $g(u)$ is a continuous function depending on $f$ defined on $[0,1]$.*

This version is from Theorem 1.1 of Chapter 17 in Lorentz et al. (1996). The KST theorem tells us that a high-dimensional continuous function can be decomposed into a composition of one-dimensional continuous functions. Thus, the analysis of high-dimensional functions can be reduced to analyzing one-dimensional functions. More references about KST can be found in He (2023); Lai & Shen (2021).

In KST, $g$ and $\phi_q(q = 1, 2, \cdots, d)$ are termed as outer and inner functions respectively. The KST structure, resembling a two-layer neural network with these functions as activation functions, prompts an interesting question: can we circumvent the curse of dimensionality in a certain function class by approximating the univariate inner and outer functions using neural networks? Noting the Lipschitz continuity of inner functions, we examine this question. If each neural network approximates a univariate Lipschitz function with an error of $\varepsilon$, each complexity is $O(\varepsilon^{-1})$. Integrating these networks into a KST structure results in an overall complexity of approximately $O(d^2)$, potentially evading the curse of dimensionality. However, unlike the inner function, the outer function lacks Lipschitz continuity. Thus, we could potentially avoid the curse of dimensionality in a function class by ensuring Lipschitz continuity of the outer function. We Define

$$K_C = \left\{ f \in C([0,1]^d) : \text{ the outer function } g_f \text{ is Lipschitz continuous with Lipschitz constant } \leq C \right\}.$$

As stated in Lai & Shen (2021), $K_C$ is dense in $C([0,1]^d)$ as $C \to \infty$. The function in $K_C$ can be represented by the form 2, where $g, \phi_q$ ($q = 1, 2, \cdots, d$) are Lipschitz continuous. Consequently, we can approximate Lipschitz continuous functions using linear spline functions and any $n$ piece linear spline can be expressed as a linear combination of $n$ ReLU functions. However, as pointed out in Lai & Shen (2021), it is not easy to judge if a function belongs to $K_C$. The outer function $g$ can vary badly even though $f$ is very smooth such as a linear polynomial. Regardless, ResNet enables us to overcome the curse of dimensionality in $K_C$. The precise result follows.

**Theorem 8.** *For any $f \in K_C$, there exists ResNet $R(x) \in \mathcal{RN}\left(d + 2, n, L\right)$ from $[0,1]^d$ to $\mathbb{R}$ with $n = \mathcal{O}(d^2\varepsilon^{-1})$ and $L = \mathcal{O}(d^2)$ such that $|R(x) - f(x)| < \varepsilon$ for any $x \in [0,1]^d$. Moreover, $R$ has $\mathcal{O}(d^4\varepsilon^{-1})$ tunable parameters.*

## 6 EXPERIMENTS

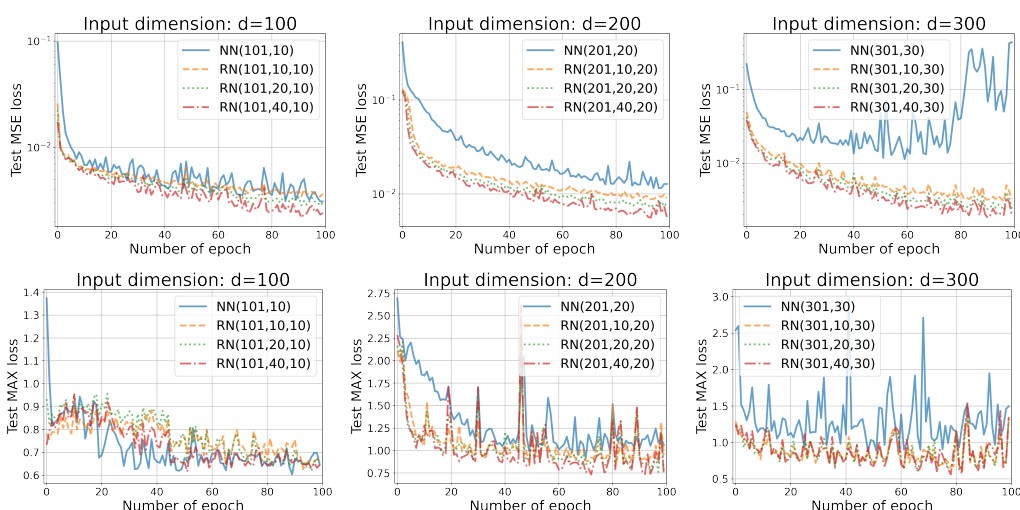

Figure 2: Comparison of testing MSE loss (top) and MAX loss (bottom).

In this section, we provide function approximation results to numerically validate the theoretical results presented in Sec. 4. Specifically, we utilize the following function to test the universal approximation capability of b-ResNet.

$$f(x) = \sum_{i=1}^{m} [a_i \prod_{j \in S_i^1} x_j + b_i \sin(\prod_{k \in S_i^2} x_k)], \tag{3}$$

where the parameter settings in equation 3 are included in Appendix H. We then compare b-ResNet with fully connected NN for approximating the function in 3 with different dimensions. The results, including the mean square error (MSE) and the infinite norm error (MAX) on testing samples, are shown in Fig. 6 and 2, respectively.

The experiment results demonstrate (i) the exceptional approximation capability of b-ResNet for learning complex functions, (ii) efficient structure design which highly reduces training parameters, and (iii) strong scalability for approximating high-dimension functions.

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

# A  NOTATIONS

In this appendix, we introduce some basic notations for use in subsequent proofs.

## A.1  FEEDFORWARD NEURAL NETWORKS (FNN)

As is known to all, FNN is a function $\Phi : \mathbb{R}^d :\to \mathbb{R}$ which is formed as the alternating compositions of ReLU function $\sigma$, and affine transformations $\mathcal{A}^{[i]}(y) = U_i y + v_i$ with $U_i \in \mathbb{R}^{d_i \times d_{i-1}}, v_i \in \mathbb{R}^{d_i}, d_0 = d$ for $i = 1, 2, \cdots, L$. Specifically,

$$\Phi\left(x\right) = \mathcal{L} \circ \sigma \circ \mathcal{A}^{[L]} \circ \sigma \circ \mathcal{A}^{[L-1]} \circ \cdots \circ \sigma \circ \mathcal{A}^{[1]}\left(x\right)$$

where $\mathcal{L}$ is a linear transformation. Here $L$ denotes the number of layers of the FNN, and the width of the FNN is conventionally defined by $\max\{d_1, d_2, \cdots, d_L\} := K$. The ReLU activation function is defined by:

$$\sigma(x) := \mathrm{ReLU}(x) = \max\left(x, 0\right) = \left(x\right)_+, x \in \mathbb{R}$$

and for $x \in \mathbb{R}^d$, $\sigma(x) := (\sigma(x_1), \cdots, \sigma(x_d))$. Typically, it is presumed that the number of neurons in each layer of an FNN is the same, which is equal to the width $K$, as any neuron deficits in a layer can be dealt with by adding $K - d_j$ neurons whose biases are zero in layer $j$. The weights between these extra neurons are consequently assigned to zero.

## A.2  RESNET

ResNet $R(x) : \mathbb{R}^d \to \mathbb{R}$ is a combination of an initial affine layer, multiple basic residual blocks with identity mapping, and a final affine output layer:

$$R(x) = \mathcal{L} \circ (\mathcal{T}^{[L]} + Id) \circ (\mathcal{T}^{[L-1]} + Id) \circ \cdots \circ (\mathcal{T}^{[1]} + Id) \circ \mathcal{A}_k(x), \qquad (4)$$

where $\mathcal{A}_k : \mathbb{R}^d \to \mathbb{R}^k$ and $\mathcal{L} : \mathbb{R}^k \to \mathbb{R}$ are affine transformations. Besides, $\mathcal{T}^{[i]}(i = 0, 1, ..., L)$ are basic residual blocks, i.e., $\mathcal{T}^{[i]}(z) = V_i \sigma\left(W_i z + b_i\right)$ where $W_i \in \mathbb{R}^{n_i \times k}, V_i \in \mathbb{R}^{k \times n_i}, b_i \in \mathbb{R}^{n_i}$.

Concretely, we denote the output of the $i$-th block by $z^{[i]}$. Then the outputs of each block can be formulated as follows:

$$\begin{aligned}
z^{[0]} &= \mathcal{A}_k\left(x\right) = W_0 x + b_0, \\
\mathcal{T}^{[i]}(z) &= V_i \sigma\left(W_i z + b_i\right), \\
z^{[i]} &= \mathcal{T}^{[i]}\left(z^{[i-1]}\right) + z^{[i-1]}, \quad i = 1, 2, \cdots, L, \\
R(x) &= \mathcal{L}\left(z^{[L]}\right) = B z^{[L]},
\end{aligned} \qquad (5)$$

where $W_0 \in \mathbb{R}^{k \times d}, b_0 \in \mathbb{R}^k, W_i \in \mathbb{R}^{n_i \times k}, V_i \in \mathbb{R}^{k \times n_i}, b_i \in \mathbb{R}^{n_i}, B \in \mathbb{R}^{1 \times k}$ and $x \in \mathbb{R}^d$.

The ResNet's depth, denoted by $L$, is defined as the number of residual blocks. The ResNet's width is the maximum number of neurons in the activation layer, that is $\max\{n_1, n_2, ..., n_L\}$. The subscript $k$ of $\mathcal{A}_k$ refers to the number of neurons in the identity layer. We denote by $\mathcal{RN}\left(k, N, L\right)$ the set of ResNet functions width $N$, depth $L$ and $k$ neurons in each identity layer.

Additionally, we define

$$\begin{aligned}
\zeta^{[i]} &= \mathcal{T}_1^{[i]}(z^{[i-1]}) = \sigma(W_i z + b_i), \\
\gamma^{[i]} &= \mathcal{T}^{[i]}\left(z^{[i-1]}\right) = \mathcal{T}_2^{[i]}\left(\zeta^{[i]}\right) = V_i \zeta^{[i]} = V_i \mathcal{T}_1^{[i]}\left(z^{[i-1]}\right) \quad \text{and} \\
z^{[i]} &= z^{[i-1]} + \gamma^{[i]}
\end{aligned} \qquad (6)$$

for $i = 1, 2, \cdots, L$. See Figure 3 for an illustration.

**Notation.** Let column vectors $a_i \in \mathbb{R}^{m_i}$, where $i = 1, 2, \cdots, n$ and $m_i \in \mathbb{N}_+ := \{1, 2, \cdots\}$. To represent these vectors concisely, we use $(a_1, a_2, \cdots, a_n)$ to denote $\left[a_1^T, a_2^T, \cdots, a_n^T\right]^T \in \mathbb{R}^{m_1 + m_2 + \cdots + m_n}$. Here, $a^T$ denotes the transpose of $a$. Let $a \in \mathbb{R}^m$. If we write a vector as $(\mathbb{R}^l, a) \in \mathbb{R}^{m+l}$, this implies that the value of the vector in the position represented by '$\mathbb{R}^l$' does not matter. If $l = 1$, we always use '-' to substitute '$\mathbb{R}$', i.e., $(-, a) \in \mathbb{R}^{m+1}$ implies the value of the position represented by '-' does not matter. For a vector $v$ in $\mathbb{R}^m$, $v_i$ is the $i$-th entry of $v$ for $i = 1, 2, \cdots, m$.

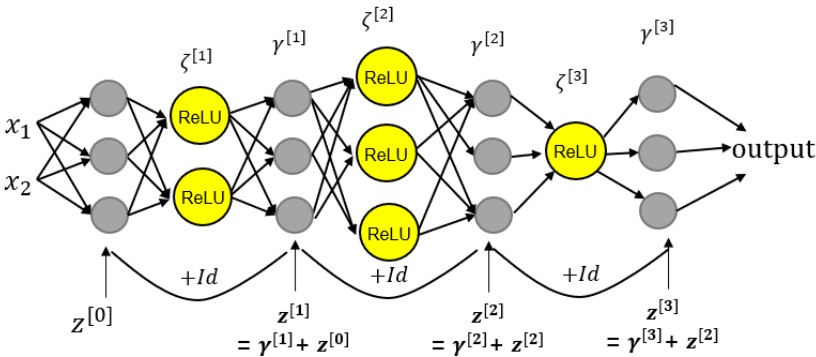

Figure 3: An illustration figure for the notation. The yellow neurons are in the activation layer and the grey neurons are in the identity layer.

# B  PROOF OF PROPOSITION 1 AND THEOREM 2

In this appendix, we provide the proofs of conclusions in Sec. 3.

## B.1  PROOF OF PROPOSITION 1

For a ResNet in $\mathcal{RN}(k, N, L)$ from $[0, 1]^d$ to $\mathbb{R}$ defined by the formula 4, 5 and 6, we now construct a special network with depth $L$ and width $N + k$ having the same output. We first suppose the input of the network is $y^{[0]} = \mathcal{A}_k(x) = z^{[0]}$ and denote the output of the $i$-th layer by $y^{[i]}$. What's more, we assume in each layer the bottom k neurons of each layer are ReLU-free, i.e., the activation function of them is identity mapping $\sigma(x) = x$. The activation of the rest of neurons are ReLU. Then by assigning some weights to the first layer, we can have $y^{[1]} = (\zeta^{[1]}, z^{[0]})$. In the next layer, we can easily compute $z^{[1]} = V_1 \zeta^{[1]} + z^{[0]}$. Then we have $y^{[2]} = (\mathbb{R}^N, z^{[1]})$. Now assume $y^{[2i]} = (\mathbb{R}^N, z^{[i]})$. Then in the first $N$ neurons of the next layer, we compute $\mathcal{T}_1^{[i+1]}(z^{[i]}) = \mathrm{ReLU}(W_{i+1} z^{[i]} + b^i)$. In the bottom $k$ ReLU-free neurons, we copy $z^{[i]}$. Then $y^{[i+1]} = (\zeta^{[i+1]}, z^{[i]})$. Then in the next layer, we can compute

$$z^{[i+1]} = V_{i+1} \zeta^{[i+1]} + z^{[i]}$$

i.e., $y^{[2(i+1)]} = (\mathbb{R}^N, z^{[i+1]})$. By induction, we have found a special $2L$ deep network with top $N$ ReLU neurons and bottom $k$ ReLU-free neurons having the same output as the ResNet. This process can be seen in Figure 4.

Next, we construct a real ReLU network $\Phi$ that has the same size and output as the special Network. Because the domain $[0, 1]^d$ is compact, there exists $C_i \in \mathbb{R}^k$ such that $z^{[i]} + C_i > 0$ for all $i = 0, 1, 2, \cdots, L$. Now, we suppose the first layer of $\Phi$ is $u^{[0]} = \mathrm{ReLU}(\mathcal{A}_k(x) + C_0) = \mathrm{ReLU}(z^{[0]} + C_0)$. Denote the $j$-th layer of $\Phi$ is $u^{[j-1]}$. In each subsequent layer of $\Phi$, the width is $N + k$. We denote $u^{[i]} = (u_{(N)}^{[i]}, u_{(k)}^{[i]})$ where $u_{(N)}^{[i]}$ is the value of top $N$ neurons and $u_{(k)}^{[i]}$ is the value of bottom $k$ neurons in the layer $i + 1$. Then note $\mathcal{T}_1^{[1]}(z^{[0]}) = \mathrm{ReLU}(W_1 z^{[0]} + b_1)$. We can compute

$$u_{(N)}^{[1]} = \mathrm{ReLU}\left(W_1(u^{[0]} - C_0) + b_1\right) = \mathrm{ReLU}(W_1 z^{[0]} + b_1) = \zeta^{[1]}$$

and $u_{(k)}^{[1]} = u^{[0]} = \mathrm{ReLU}(z^{[0]} + C_0)$. Note $z^{[i]} = \mathrm{ReLU}(z^{[i]} - C_i) + C_i$. We can compute

$$u_{(k)}^{[2]} = \mathrm{ReLU}\left(V_1 u_{(N)}^{[1]} + u_{(k)}^{[1]} - C_0 + C_1\right) = \mathrm{ReLU}\left(V_1 \zeta^{[1]} + z^{[0]} + C_1\right) = \mathrm{ReLU}(z^{[1]} + C_1).$$

Now, we suppose $u^{[2j]} = (\mathbb{R}^N, \mathrm{ReLU}(z^{[j]} + C_j))$. Then we can compute

$$u_{(N)}^{[2j+1]} = \mathrm{ReLU}\left(W_{j+1}(u^{[2j]} - C_j) + b_{j+1}\right) = \mathrm{ReLU}(W_{j+1} z^{[j]} + b_{j+1}) = \mathcal{T}_1^{[j+1]}(z^{[j]}) = \zeta^{[j+1]}$$

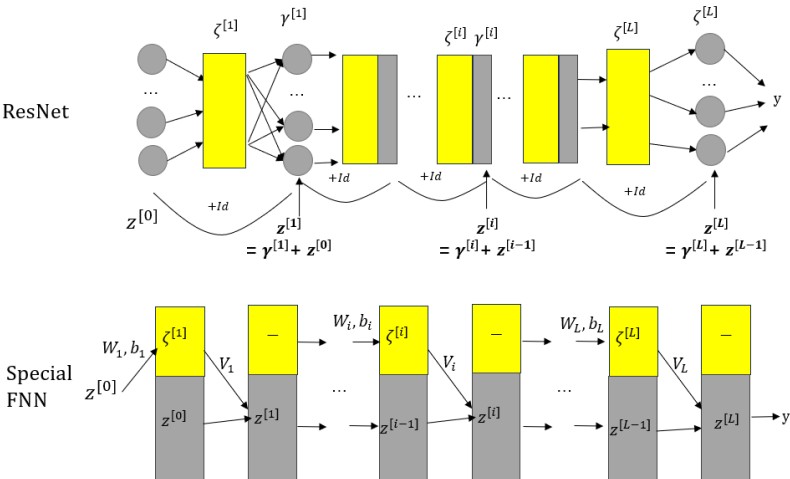

Figure 4: ResNet (top) can be generated by a special FNN (bottom). All grey neurons are ReLU-free and all yellow neurons are with ReLU activation. Moreover, for ResNet, the yellow neurons are in the activation layer and the grey neurons are in the identity layer.

and $u_{(k)}^{[j+1]} = \text{ReLU}(u_{(k)}^{[j]}) = \text{ReLU}(z^{[0]} + C_0)$. Then in the next layer,

$$
\begin{aligned}
u_{(k)}^{[2j+2]} &= \text{ReLU}\left(V_{j+1}u_{(N)}^{[2j+1]} + u_{(k)}^{[2j+1]} - C_j + C_{j+1}\right) \\
&= \text{ReLU}\left(V_{j+1}\mathcal{T}_1^{[j+1]}(z^{[j]}) + z^{[j]} + C_{j+1}\right) \\
&= \text{ReLU}(z^{[j+1]} + C_{j+1}).
\end{aligned}
$$

By induction, we can output $z^{[L]}$ in the last layer, i.e., $u^{[2L]} = (\mathbb{R}^N, \text{ReLU}(z^{[L]} + C_L))$. Then output $z^{[L]}$ by some affine transformation $\mathcal{A}(u^{[2L]}) = \text{ReLU}(z^{[L]} + C_L) - C_L = z^{[L]}$.

From the construction of the ReLU network, we can see the FNN has $\Theta(W + kL)$ non-zero training weights.

## B.2 PROOF OF THEOREM 2

Because the product function $f(x) = x_1x_2\cdots x_d$ belongs to $\mathcal{P}(d,p)$, it suffices to show the lower bound of the complexity of an FNN on the approximation of $f$ is $\Theta_d(\log 1/\varepsilon)$. Let $x \in [0,1]$. Define $\widetilde{\Psi}(x) = \Psi(x,x,\cdots,x)$ and $\widetilde{f}(x) = f(x,x,\cdots,x) = x^d$. Then by the assumption, we have

$$
\left|\widetilde{\Psi}(x) - \widetilde{f}(x)\right| < \varepsilon, \quad x \in [\tfrac{1}{2}, 1].
$$

In the interval $[1/2, 1]$, $\widetilde{f}$ is strictly convex because

$$
\widetilde{f}''(x) = d(d-1)x^d \ge d(d-1)(\tfrac{1}{2})^d := c_1 > 0.
$$

By lemma 2.1 in Telgarsky (2015), $\widetilde{\Psi}$ is a CPwL function over $[0,1]$ with at most $(2N)^L$ linear pieces, i.e. $[1/2, 1]$ is partitioned into at most $(2N)^L$ intervals for which $\widetilde{\Psi}$ is linear. Now, we divide $[1/2, 1]$ into $(2N)^L$ intervals. Thus, there exists an interval $[a,b] \subset [1/2, 1]$ with $b - a \ge \frac{1}{2(2N)^L}$ over which $\widetilde{\Psi}$ is linear. Then define

$$
G(x) = \widetilde{f}(x) - \widetilde{\Psi}, x \in [a,b].
$$

Then $|G(x)| < \varepsilon$ and $G''(x) \ge c_1 > 0$ for any $x \in [a,b]$ due to the linearity of $\widetilde{f}$. Then we consider $x \in [a,b]$ and local taylor expansion at $(a+b)/2$:

$$
G(x) = G(\tfrac{a+b}{2}) + G'(\tfrac{a+b}{2})(x - \tfrac{a+b}{2}) + \frac{G''}{2}(\xi)(x - \tfrac{a+b}{2})^2 \quad \text{where} \xi \in [a,b].
$$

Then let $x = a$ and $x = b$, we have

$$G(a) = G(\frac{a+b}{2}) - G'(\frac{a+b}{2})(\frac{b-a}{2}) + \frac{G''}{2}(\xi)(\frac{b-a}{2})^2, \quad \xi \in [a, \frac{a+b}{2}] \quad \text{and}$$

$$G(b) = G(\frac{a+b}{2}) + G'(\frac{a+b}{2})(\frac{b-a}{2}) + \frac{G''}{2}(\eta)(\frac{b-a}{2})^2, \quad \eta \in [\frac{a+b}{2}, b].$$

It follows that

$$\max\{G(a), G(b)\} \geq G(\frac{a+b}{2}) + \frac{c_1}{2}(\frac{b-a}{2})^2.$$

Thus, by noting $b - a \geq \frac{1}{2(2N)^L}$ we have

$$2\varepsilon > \max\{G(a), G(b)\} - G(\frac{a+b}{2}) \geq \frac{c_1}{2}(\frac{b-a}{2})^2 \geq \frac{c_1}{2}\left(\frac{1}{4(2N)^L}\right)^2.$$

Then

$$(2N)^{2L} \geq \frac{c}{\varepsilon}$$

where $c$ is a constant depending on $d$. It follows from the number of neurons $T = NL$ that

$$\log \frac{2T}{L} \geq \frac{1}{2L} \log \frac{c}{\varepsilon}$$

$$\iff 4T \geq \frac{u}{\log u} \log \frac{c}{\varepsilon} \geq \log \frac{c}{\varepsilon} \quad \text{where} \quad u = \frac{2T}{L}.$$

Therefore, the number of neurons must be at least the order $\log 1/\varepsilon$.

## C    PROOF OF THEOREM 3 AND THEOREM 4

Our ideas in this appendix are from DeVore et al. (2021). It should be noted that while DeVore et al. (2021) inspired our approach to polynomial approximation, there are big differences in the construction details. Significantly, our main contribution is the successful demonstration of ResNet's construction proof.

The discussion in Subsection C.1 commences with a consideration of the fundamental functions, ranging from $x^2$ to $xy$, which we use to construct our approximation using ResNet. Subsequently, in Subsection C.2, we begin by establishing Theorem 3 for the case $[0, 1]^d$. This is then extended to the case $[-M, M]^d$ for $M > 1$ in Subsection C.3.

### C.1    PRELININARIES

We recall that the so-called hat function $h$ is defined by

$$h(x) = 2(x)_+ - 4\left(x - \frac{1}{2}\right)_+ + 2(x - 1)_+. \tag{7}$$

Let $h_m(x)$ be the $m$-fold composition of the function $h$, i.e. $h_m = \underbrace{h \circ h \circ \cdots \circ h}_{m \text{ times}}$ which is the so-called sawtooth function. Then

$$x^2 = x - \sum_{m \geq 1} 4^{-m} h_m(x), \quad x \in [0, 1].$$

Next, we define

$$S(x) := x^2 \quad \text{and} \quad S_n(x) := x - \sum_{m=1}^{n} 4^{-m} h_m(x), \quad n \geq 1, \quad x \in [0, 1].$$

We then have

$$|S(x) - S_n(x)| \leq \sum_{k=n+1}^{\infty} 4^{-k} \leq \frac{1}{3} \cdot 4^{-n}, \quad x \in [0, 1]. \tag{8}$$

$S_n(x)$ is a piecewise linear interpolation of $S$ on $[0, 1]$, using $2^n + 1$ uniformly distributed breakpoints, as indicated in Yarotsky (2017) (see Proposition 1). Using equation 8, we can generate $S_n$ and approximate $x^2$.

**Proposition 9.** *There exists a ResNet $R(x) \in \mathcal{RN}(2, 4, L)$ with $L = \mathcal{O}(\log 1/\varepsilon)$ such that*

$$\left| R(x) - x^2 \right| < \varepsilon, x \in [0, 1]$$

*while having $\mathcal{O}(\log 1/\varepsilon)$ neurons. Especially, $R(x) \in \mathcal{RN}(4, 2, n)$ generate $S_n$ exactly.*

*Proof.* It suffices to construct a ResNet required to represent $S_n$. Then we let the right-hand side of (8) equal to $\varepsilon$, i.e., $\frac{1}{3}4^{-n} = \varepsilon$. We then have $n = \mathcal{O}(\log 1/\varepsilon)$. Next, we construct a ResNet $R(x) \in \mathcal{RN}(4, 2, n)$ generating $S_n(x)$ exactly.

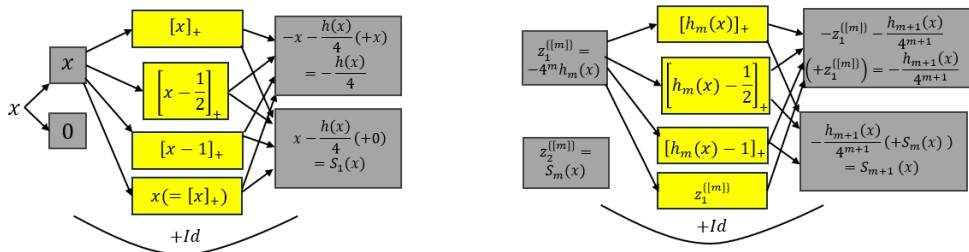

Figure 5: Illustration for the constructive first block (left) and the $m$-th block (right). Grey represents the identity layer and yellow represents the activation layer.

Let $z^{[0]} = \mathcal{A}_2(x) = (x, 0)$. Then by equation (7), we can use three ReLU units to store $(x)_+$, $(x - 1/2)_+$ and $(x - 1)_+$ and hence compute $h(x)$. At the same layer, use one ReLU unit to copy $x$. Then in the first residual block, we can compute

$$\gamma^{[1]} = \mathcal{T}^{[1]}(z^{[0]}) = (-x - h(x)/4, x - h(x)/4)$$

so that

$$z^{[1]} = \gamma^{[1]} + z^{[0]} = (-x - h(x)/4, x - h(x)/4) + (x, 0) = (-h(x)/4, S_1(x)).$$

See Figure 5 (left) for illustration.

Now we assume the output of the $m$-th block is $z^{[m]} = (-4^m h_m(x), S_m(x))$. Also by (7) and assigning appropriate weights, we can use three units to compute $h(h_m(x)) = h_{m+1}(x)$ and use one unit to copy $-4^m h_m(x)$ by $a = -(-a)_+$. Thus, we can output

$$\gamma^{[m+1]} = \mathcal{T}^{[m+1]}(z^{[m]}) = (-4^{m+1}h_{m+1}(x) - 4^m h_m(x), -4^{m+1}h_{m+1}(x))$$

in the next block so that

$$z^{[m+1]} = (-4^{m+1}h_{m+1}(x) - 4^m h_m(x), -4^{m+1}h_{m+1}(x)) + z^{[m]} = (-4^{m+1}h_{m+1}(x), S_{m+1}(x)).$$

See Figure 5 (right) for illustration. By induction, we complete our proof. Concretely, $z^{[n]} = (-4^n h_n(x), S_n(x))$ and $R(x) = \mathcal{L}(z^{[n]}) = S_n(x)$ by letting $\mathcal{L}(x_1, x_2) = x_2$. $\square$

Let $x, y \in [0, 1]$. We can approximate the product function $xy$ by using the equality $xy = (\frac{|x+y|}{2})^2 - (\frac{|x-y|}{2})^2$. Define

$$\pi_n(x, y) = S_n(\frac{|x+y|}{2}) - S_n(\frac{|x-y|}{2}). \tag{9}$$

It then follows from equation (8),

$$|\pi_n(x, y) - xy| \leq 4^{-n}, \quad \forall x, y \in [0, 1]. \tag{10}$$

For the later rigorous derivation, we also need to prove the following lemma:

**Lemma 10.**

$$\pi_n(x, y) \in [0, 1], \quad \forall x, y \in [0, 1]. \tag{11}$$

*Proof.* According to the definition of $S_n$, we have

$$x^2 \le S_n(x) \le x, \quad \text{for} \quad x \in [0, 1].$$

Then

$$\begin{aligned}
\pi_n(x, y) &= S_n(\frac{x+y}{2}) - S_n(\frac{|x-y|}{2}) \\
&\le \frac{x+y}{2} - \frac{(x-y)^2}{4} \\
&= \frac{1}{4}\left(x(2-x) + y(2-y) + 2xy\right) \\
&\le \frac{1}{4}(1 + 1 + 2) = 1.
\end{aligned}$$

Next, we show $\pi_n(x, y) \ge 0$. We start from

$$\begin{aligned}
\pi_n(x, y) &= S_n(\frac{x+y}{2}) - S_n(\frac{|x-y|}{2}) \\
&= \frac{x+y}{2} - \sum_{i=1}^{n} 4^{-i} h_i(\frac{x+y}{2}) - \frac{|x-y|}{2} + \sum_{i=1}^{n} 4^{-i} h_i(\frac{|x-y|}{2}) \\
&= \min\{x, y\} + \sum_{i=1}^{n} 4^{-i}\left(h_i(\frac{|x-y|}{2}) - h_i(\frac{x+y}{2})\right).
\end{aligned} \tag{12}$$

Now we introduce the function

$$\zeta(x) := 2\min\{|x - s| : s \in \mathbb{Z}\}, \quad x \in \mathbb{R}.$$

Then for $x \in [0, 1]$ we have

$$h(x) = \zeta(x) \quad \text{and} \quad h_m(x) = \zeta(2^{m-1}x), m \ge 2.$$

Since $\zeta$ is subadditive, i.e. $\zeta(t + t') \le \zeta(t) + \zeta(t')$, we have

$$\begin{aligned}
h_i(\frac{x+y}{2}) &= h_i(\frac{|x-y|}{2} + \min\{x, y\}) \\
&= \zeta(2^{i-1}(\frac{|x-y|}{2} + \min\{x, y\})) \\
&\le \zeta(2^{i-1}\frac{|x-y|}{2}) + \zeta(2^{i-1}\min\{x, y\}) \\
&= h_i(\frac{|x-y|}{2}) + h_i(\min\{x, y\}).
\end{aligned} \tag{13}$$

Namely,

$$h_i(\frac{|x-y|}{2}) - h_i(\frac{x+y}{2}) \ge -h_i(\min\{x, y\}).$$

From (12), we then have

$$\pi_n(x, y) \ge \min\{x, y\} - \sum_{i=1}^{n} 4^{-i}(\min\{x, y\}) = S_n(\min\{x, y\}) \ge \min\{x, y\}^2 \ge 0.$$

$\square$

For $x, y \in [-M, M]$, we can approximate $xy$ by the following remark.

**Remark 11.** *If $x, y \in [-M, M]$, we can approximate $xy$ by using the equality $xy = M^2\left[(\frac{|x+y|}{2M})^2 - (\frac{|x-y|}{2M})^2\right]$. Because the domain of $S_n$ is $[0, 1]$, we define*

$$\widehat{\pi}_n(x, y) = M^2\pi_n(\frac{x}{M}, \frac{y}{M}) = M^2\left[S_n(\frac{|x+y|}{2M}) - S_n(\frac{|x-y|}{2M})\right]. \tag{14}$$

*We have*

$$|\widehat{\pi}_n(x, y) - xy| \le M^2 4^{-n}, \forall x, y \in [-M, M]. \tag{15}$$

Now we show ResNet can approximate the product function $xy$.

**Proposition 12.** *Let $x, y \in [-M, M]$. There exists a ResNet*

$$R \in \mathcal{RN}\left(3, (4, L)\right)$$

*from $[-M, M]$ to $\mathbb{R}$ with $L = \mathcal{O}(\log M/\varepsilon)$ such that*

$$|R(x, y) - xy| < \varepsilon, x, y \in [-M, M]$$

*while having $\mathcal{O}(\log M/\varepsilon)$ neurons and tunable weights. Especially, the ResNet $R(x, y)$ with width $4$ and depth $2n$ can generate $\pi_n(x, y)$ exactly.*

*Proof.* It suffices to construct a ResNet required to output $\pi_n(x, y)$. Let the right-hand side of (15) equal $\varepsilon$. We can get $n = \mathcal{O}(\log M/\varepsilon)$. Now, we construct a ResNet with width $4$ and depth $2n$ to represent $\pi_n(x, y)$.

Let $\mathcal{A}_3(x) = (\frac{x+y}{2M}, \frac{x-y}{2M}, 0)$. Next, we can use the first block to output $z^{[1]} = (\frac{|x+y|}{2M}, \frac{|x-y|}{2M}, 0)$ by the simple observation $|a| = (a)_+ + (-a)_+$. See Figure 6 for illustration.

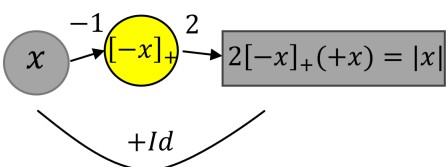

Figure 6: Illustration for computing $|x|$ by a residual block.

Then, it follows from the proof of 9 that we can use the first $n$ layers and $4$ units in each layer to output $z^{[n+1]} = (-, \frac{|x-y|}{2M}, S_n(\frac{|x+y|}{2M}))$ while keeping the value of the second neuron in each identity layer unchanged. Next, by the same operation, we use the next $n$ blocks to output

$$z^{[2n+1]} = (-, -, S_n(\frac{x+y}{2}) - S_n(\frac{|x-y|}{2})) = (-, -, \pi_n(x, y)/M^2).$$

Thus, $R(x, y) = \mathcal{L}(-, -, \pi_n(x, y)/M^2) = \pi_n(x, y)$ by letting $\mathcal{L}(x_1, x_2, x_3) = M^2 x_3$. See Figure 7 for illustration.

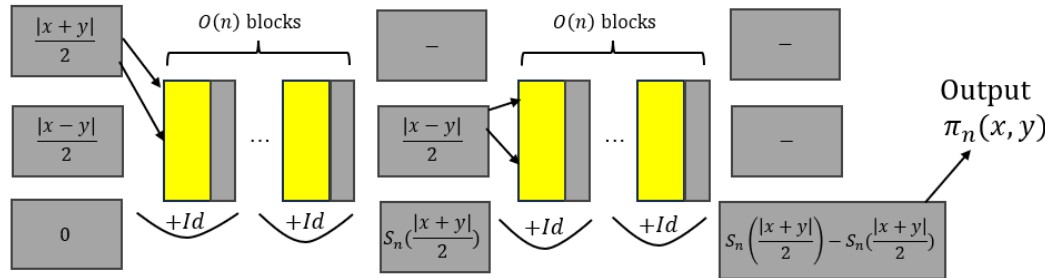

Figure 7: Illustration for generating $\pi_n(x, y)$ by the constructive ResNet. Grey represents the identity layer and yellow represents the activation layer.

$\square$

Moreover, we can approximate the multiple product function $x_1 x_2 \cdots x_d$ where $x_1, x_2, \cdots, x_d \in [0, 1]$. We can well-define by (11) that

$$\pi_n^m(x_1, x_2, \cdots, x_m) = \pi_n\left(\pi_n^{m-1}(x_1, x_2, \cdots, x_{m-1}), x_m\right), m = 3, 4, \cdots \quad \text{for} \quad x_1, \cdots, x_d \in [0, 1] \tag{16}$$

and $\pi_n^2(x_1, x_2) = \pi_n(x_1, x_2)$. Then we have

**Proposition 13.**

$$|\pi_n^m(x_1, x_2, \cdots, x_m) - x_1 x_2 \cdots x_m| \le em4^{-n}, \quad x_1, x_2, \cdots, x_m \in [0,1] \qquad (17)$$

*as long as* $n \ge 1 + \log_2 m$.

*Proof.* First, It follows from the definition of $S_n$ and $S(x) = x^2$ that

$$S(x) - S_n(x) = -\sum_{m=n+1}^{\infty} 4^{-m} h_m(x).$$

Note $h_m$ has the Lipschitz norm $2^m$. We have

$$\|S' - S_n'\|_{L_\infty([0,1])} \le |\sum_{m=n+1}^{\infty} 4^{-m}| \cdot \|h_m'(x)\|_{L_\infty([0,1])}$$

$$\le |\sum_{m=n+1}^{\infty} 4^{-m} 2^m|$$

$$\le 2^{-n}, \quad n \ge 1.$$

$S'(x) = 2x$ so we have

$$S_n'(x) = 2x + \delta \quad \text{where} \quad \delta \le 2^{-n}. \qquad (18)$$

Define $\pi(x, y) := xy$. We have

$$\partial_1 \pi_n(x, y) = S_n'\left(\frac{x+y}{2}\right) - \frac{1}{2} \cdot 1_{\{x \ge y\}} S_n'\left(\frac{x-y}{2}\right) + \frac{1}{2} \cdot 1_{\{x < y\}} S_n'\left(\frac{y-x}{2}\right)$$

$$= \frac{1}{2}(x + y + \delta) - \frac{1}{2} \cdot 1_{\{x \ge y\}}(x - y + \delta_1) + \frac{1}{2} \cdot 1_{\{x < y\}}(y - x + \delta_2) \qquad (19)$$

$$= y + \delta - 1_{\{x \ge y\}} \delta_1 + 1_{\{x < y\}} \delta_2$$

$$= \partial_1 \pi(x, y) + \delta - 1_{\{x \ge y\}} \delta_1 + 1_{\{x < y\}} \delta_2,$$

where $\delta, \delta_1, \delta_2 \le 2^{-n}$. Moreover, it is the same when considering about $\partial_2 \pi_n(x, y)$. Thus, we have

$$\|\partial_i \pi - \partial_i \pi_n\|_{L_\infty([0,1]^2)} \le 2^{-n+1}, \quad \text{where} \quad \partial_i := \partial_{x_i}, i = 1, 2. \qquad (20)$$

Define $\pi^m(x_1, x_2, \cdots, x_m) = x_1 x_2 \cdots x_m$. Now we assume

$$\|\pi^j - \pi_n^j\|_{C([0,1]^j)} \le c_j 4^{-n} \qquad (21)$$

holds for all $j \le m - 1$. Note, the inequality literally holds for $j = 2$ by letting $c_2 = 1$. Then we have

$$\|\pi^m - \pi_n^m\|_{C([0,1]^m)} \le \|\pi(x_m, \pi^{m-1}) - \pi_n(x_m, \pi^{m-1})\|_{C([0,1]^m)} + \|\pi_n(x_m, \pi^{m-1}) - \pi_n(x_m, \pi_n^{m-1})\|_{C([0,1]^m)}$$

$$\le 4^{-n} + \|\partial_2 \pi_n\|_{L_\infty([0,1]^2)} \|\pi^{m-1} - \pi_n^{m-1}\|_{C([0,1]^{m-1})}$$

$$\le 4^{-n} + (1 + 2^{-n+1} c_{m-1}) 4^{-n}$$

$$= (1 + \alpha_n c_{m-1}) \cdot 4^{-n}.$$

By induction, we have proved

$$\|\pi^m - \pi_n^m\|_{C([0,1]^m)} \le c_m 4^{-n}$$

where $c_m$ satisfies the recurrence formula $c_m = 1 + \alpha_n c_{m-1}, m \ge 3$, with initial value $C_2 = 1$. The solution is

$$c_m = \sum_{j=0}^{m-2} \alpha_n^j \le (m-1) \alpha_n^{m-2}.$$

If $m \le 2^{n-1}$, we have

$$c_m \le (m-1)\left(1 + \frac{1}{2^{n-1}}\right)^{m-2} < m\left(1 + \frac{1}{m}\right)^m < em.$$

Then we complete the proof and get

$$\|\pi^m - \pi_n^m\|_{C([0,1]^m)} \le em4^{-n}$$

for $m = 3, 4, 5, \cdots$, as long as $n \ge 1 + \log_2 m$. $\qquad \square$

C.2 PROOF OF THEOREM 3 OVER $[0, 1]^d$

Now, we are ready to prove Theorem 3 over $[0, 1]^d$. For the completeness, we show the following theorem.

**Theorem 14** (Theorem 3). *Let* $x = (x_1, x_2, \cdots, x_d) \in [0, 1]^d, \alpha \in \mathbb{N}^d$ *and* $x^\alpha$ *be any given monomial with degree* $p$. *Then*

*(1) there is a ResNet* $R_1 \in \mathcal{RN}(d + 1, 4, \mathcal{O}(d \log(d/\varepsilon)))$ *such that*
$$\|R_1 - x_1 x_2 \cdots x_d\|_{C([0,1]^d)} < \varepsilon$$
*while having* $\mathcal{O}(d \log(d/\varepsilon))$ *tunable weights. Moreover, there is a ResNet belonging to* $R_1 \in \mathcal{RN}(d + 1, 4, \mathcal{O}(nd))$ *can generate* $\pi_n^d(x_1, x_2, \cdots, x_3)$ *exactly.*

*(2) there is a ResNet* $R_2 \in \mathcal{RN}(d + 3, 4, \mathcal{O}(p \log(p/\varepsilon)))$ *such that*
$$\|R_2 - x^\alpha\|_{C([0,1]^d)} < \varepsilon$$
*while having* $\mathcal{O}(p \log(p/\varepsilon))$ *tunable weights.*

*Proof.* We prove (1) first. Let the right-hand side of inequality 17 equal to $\varepsilon$ and note $m = d$ under the condition of (1). Then, $n$ is the order $\log d/\varepsilon$. Now we construct a ResNet required with width 4 and depth $\mathcal{O}(nd)$ generating $\pi_n^d(x_1, x_2, \cdots, x_d)$.

Let $\mathcal{A}_{d+1} = (x_1, x_2, \cdots, x_d, 0)$. It follows from the proof of proposition 12 that we can assign some weights for the first $2n$ blocks to output
$$z^{[2n]} = (-, -, x_3, x_4, \cdots, x_d, \pi_n(x_1, x_2))$$
while only changing the value of the first, second, and last neurons in each activation layer. In the next block, we set the value of the first and second neurons to zero by using identity mapping. In the next block, we then can output
$$z^{[2n+1]} = (0, 0, x_3, x_4, \cdots, x_d, \pi_n(x_1, x_2)).$$
The zero-value neuron in the activation layer is ready to store the results in the next phase. Then in the next $2n$ blocks we can compute $\pi_n^3(x_1, x_2, x_3)$. Concretely, by the proof of proposition 12, we can have
$$z^{[4n+1]} = (0, \pi_n^3(x_1, x_2, x_3), -, x_4, \cdots, x_d, -).$$
By repeatedly doing the operation above, we can use about $(2n+1)(d-1)$ blocks totally with width 4 to approximate $\pi_n^d(x_1, x_2, \cdots, x_d)$. Moreover, there are about $4d + 4$ weights in each building block. However, from the operation above, we can see only a constant number of weights are non-zero. Therefore, this network has at most $cdn$ tunable weights where $c$ is an absolute constant.

For (2), if $p \le d$, the case can be the same with (1). Let's assume $p > d$. We just need to note $x_1^{\alpha_1} x_2^{\alpha_2} \cdots x_d^{\alpha_d}$ can be approximated by
$$\pi_n^d(\pi_n^{\alpha_1}(x_1, \cdots, x_1), \cdots, \pi_n^{\alpha_d}(x_d, \cdots, x_d)) = \pi_n^p(\underbrace{x_1, \cdots, x_1}_{\alpha_1 \text{ times}}, \underbrace{x_2, \cdots, x_2}_{\alpha_2 \text{ times}}, \cdots, \underbrace{\alpha_d, \cdots, \alpha_d}_{\alpha_d \text{ times}}).$$
(22)

Thus, we must store the value of $x_1, \cdots, x_d$ in each building block. However, in the proof of (1), if we complete the output of $\pi_n(x_1, x_2)$, we will lose the value of $x_1$ and $x_2$ in the neurons. That's why we need $d + 3$ neurons in the identity layer in this case. The two more neurons in the identity layer can help us preserve $x_1, \cdots, x_d$ in each identity layer. Here we briefly show a constructive ResNet generating 22 exactly.

Let $\mathcal{A}_{d+3} = (x, x_1, x_1, 0)$. By the proof of proposition 12, we can compute $\pi_n(x_1, x_1)$ using $2n$ blocks and get $z^{[2n]} = (x, -, -, \pi_n(x_1, x_1))$ in the $2n$-th block. In the next block, we compute $z^{[2n+1]} = (x, 0, x_1, \pi_n(x_1, x_1))$. Then in the next $2n$ block, we compute $\pi_n^3(x_1, x_1, x_1) = \pi_n^2(x_1, \pi_n(x_1, x_1))$ and hence get $z^{[4n+1]} = (x, \pi_n^3(x_1, x_1, x_1), -, -)$. Then by doing it repeatedly, we can use $\mathcal{O}(pn)$ blocks to generate (22). Similarly by the inequality (17) where $m = p$, we can get the depth is $\mathcal{O}(p \log \frac{p}{\varepsilon})$ if the desired accuracy is $\varepsilon$. Moreover, note there is only a constant number of weights being non-zero in each block. Thus, the total number of the non-zero weights (tunable weights) is $\mathcal{O}(p \log \frac{p}{\varepsilon})$. We can see an illustration in Figure 8.

$\square$

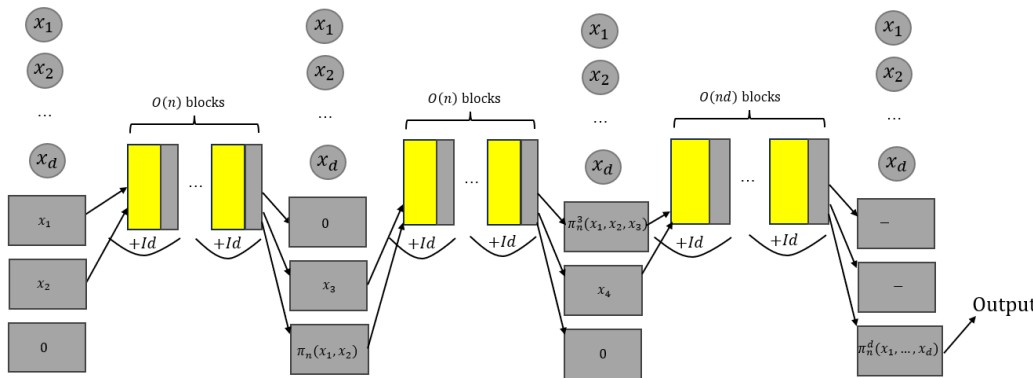

Figure 8: Illustration for generating $\pi_n^d(x_1, \cdots, x_d)$ by the constructive ResNet. The topmost $d$ neurons are used for storing values. Grey represents the identity layer and yellow represents the activation layer.

Following this theorem, we supplement some discussions about **why are monomials important.** Monomials are the essential constituents of polynomials, which serve an integral role in both theory and applications. Additionally, monomials possess a straightforward mathematical structure, which aids in analyzing and comparing the approximation capabilities of neural networks. Numerous intriguing studies have been conducted in this field. Both deep Yarotsky (2017) and shallow Blanchard & Bennouna (2021) ReLU networks can efficiently approximate monomials over $[0, 1]^d$ with $poly(d)$ neurons. However, for monomials over $[0, M]^d$ ($M > 1$), shallow ResLU networks will at least cost $exp(d)$ neurons approximating it to $\varepsilon$ Shapira (2023). Further, the cost of shallow networks will be reduced to $poly(d)$ if the monomial is normalized over $[0, M]^d$ (multiplied by some normalization constant $M^{-p}$). More comprehensive discussion can be found in Shapira (2023).

### C.3 PROOF OF THEOREM 3

In this subsection, we extend Thm, 14 to $[-M, M]^d$ where $M > 1$. First, we extend the lemma 10 to the $[-1, 1]$.

**Lemma 15.** *Let $x, y \in [-1, 1]$. Then $\pi_n(x, y) \in [-1, 1]$*

*Proof.* Since $x, y \in [-1, 1]$, it suffices to show

$$|\pi_n(x, y)| = \left| S_n(\frac{|x + y|}{2}) - S_n(\frac{|x - y|}{2}) \right| \leq 1. \tag{23}$$

First, we show that $S_n(x)$ is monotone incresing over $[0, 1]$. For any $0 \leq x \leq y \leq 1$,

$$S_n(x) - S_n(y) = x - y + \sum_{i=1}^{n} 4^{-i} \left( h_i(y) - h_i(x) \right).$$

Then by

$$h_i(x) = h_i(y + x - y) = \zeta(2^{i-1}(y + x - y)) \leq \zeta(2^{i-1}y) + \zeta(2^{i-1}(x - y))) = h_i(y) + h_i(x - y)$$

we have

$$S_n(x) - S_n(y) = x - y + \sum_{i=1}^{n} 4^{-i} \left( h_i(y) - h_i(x) \right) \geq x - y - \sum_{i=1}^{n} 4^{-i} h_i(x - y) \geq (x - y)^2 \geq 0.$$

Note $|x + y| \geq |x - y|$ is equivalent to $xy \geq 0$. So we only care about the following case to show 23:

- $x, y \le 0$.

$$\pi_n(x, y) = S_n(\frac{|x+y|}{2}) - S_n(\frac{|x-y|}{2})$$
$$\le \frac{|x|+|y|}{2} - \frac{(x-y)^2}{4}$$
$$= \frac{1}{4}\left(|x|(2-|x|) + |y|(2-|y|) + 2xy\right)$$
$$\le \frac{1}{4}(1+1+2) = 1.$$

- $xy \le 0$.

$$\pi_n(x, y) = S_n(\frac{|x-y|}{2}) - S_n(\frac{|x+y|}{2})$$
$$\le \frac{|x|+|y|}{2} - \frac{(x+y)^2}{4}$$
$$= \frac{1}{4}\left(|x|(2-|x|) + |y|(2-|y|) - 2xy\right)$$
$$\le \frac{1}{4}(1+1+2) = 1.$$

$\square$

Thus, $\pi_n^m$ can be well-defined over $[-1, 1]$ (equation 16). Next, we show that Proposition 13 holds for $x_1, x_2, \cdots, x_d \in [-1, 1]$, i.e.,

**Proposition 16.**

$$|\pi_n^m(x_1, x_2, \cdots, x_m) - x_1 x_2 \cdots x_m| \le em4^{-n}, \quad x_1, x_2, \cdots, x_m \in [-1, 1] \tag{24}$$

*as long as $n \ge 1 + \log_2 m$.*

*Proof.* For $x, y \in [-1, 1]$, the only change of the proof is equation 19. We note

$$S_n'(\frac{|x+y|}{2}) = 1_{\{x+y \ge 0\}} \frac{1}{2}(x+y+\varepsilon_1) - 1_{\{x+y<0\}} \frac{1}{2}(-x-y+\varepsilon_2) = \frac{1}{2}(x+y+\varepsilon)$$

where $\varepsilon_1, \varepsilon_2, \varepsilon \le 2^{-n}$. Then we can still get

$$\|\partial_i \pi - \partial_i \pi_n\|_{L_\infty([-1,1]^2)} \le 2^{-n+1}, \quad \text{where } \partial_i := \partial_{x_i}, i = 1, 2. \tag{25}$$

Then we can show the result following the proof of C.1. $\square$

Then Thm. 3 can be easily showed by the following remark.

**Remark 17** (Theorem 3). *Let $x_1, x_2, \cdots, x_m \in [-M, M]$. To approximate $x_1 x_2 \cdots x_m$, we consider a function defined by $\widehat{\pi}_n^m(x_1, \ldots, x_m) := M^m \pi_n^m(|x_1|/M, \ldots, |x_m|/M)$ with the approximation accuracy*

$$|x_1 x_2 \cdots x_m - \widehat{\pi}_n^m(x_1, \cdots, x_m)| \le em M^m \cdot 4^{-n}, \quad \forall x_1, x_2, \cdots, x_m \in [-M, M] \tag{26}$$

*as long as $n \ge 1 + \log_2 m$. Moreover $\widehat{\pi}_n^m$ can be generated by a ResNet $R(x) \in \mathcal{RN}(m+1, 4, \mathcal{O}(mn))$ while having at most $\mathcal{O}(mn)$ tunable weights.*

*Proof.* The remark is the direct corollary from the proof of Theorem 14. By letting the right hand side of Equation 26 equal to $\varepsilon$ where $m = p$ and $p$ is the degree of the monomial, we complete the proof of Theorem 3. $\square$

### C.4 PROOF OF THEOREM 4

*Proof.* By Theorem 3, for each $x^\alpha : \alpha \in E$, we can use a ResNet with width 4, depth $\mathcal{O}(p \log p/\varepsilon)$ and $d + 3$ neurons in each identity layer to output $R_\alpha(x)$ such that

$$|R_\alpha(x) - x^\alpha| < \varepsilon, \quad x \in [0, 1]^d.$$

Thus,

$$|\sum_{\alpha \in E} c_\alpha R_\alpha(x) - \sum_{\alpha \in E} c_\alpha x^\alpha| < \varepsilon \cdot \sum_{|\alpha| \in E} |c_\alpha|, \quad x \in [0, 1]^d.$$

Then Let $\mathcal{A}_{d+4} = (x, x_1, x_1, 0, 0)$. To generate each $R_\alpha(x)$, we need 4 more computational units in each identity layer and depth $\mathcal{O}(p \log p/\varepsilon)$ while having $\mathcal{O}(p \log p/\varepsilon)$ non-zero weights. Then store the value of $R_\alpha(x)$ in the last neuron in each identity layer. Then we can output $(x, -, -, -, \sum_{\alpha \in E} c_\alpha R_\alpha(x))$ finally with depth $\mathcal{O}(p|E| \log(p/\varepsilon))$ while having $\mathcal{O}(p|E| \log(p/\varepsilon))$ non-zero weights totally. □

## D PROOF OF THEOREM 5

In this section, we prove Theorem 5. Before that, we will give the definition supplement about Sobolev space in subsection 4.2.

### D.1 DEFINITION SUPPLEMENT OF SOBOLEV SPACES

For $\alpha = (\alpha_1, \ldots, \alpha_d) \in \mathbb{N}^d$ and $x = (x_1, \ldots, x_d) \in [0, 1]^d$, define

$$D^\alpha f = \frac{\partial^{|\alpha|} f}{\partial x_1^{\alpha_1} \cdots \partial x_d^{\alpha_d}}$$

where $|\alpha| = \alpha_1 + \cdots + \alpha_d$. Let $r \in \mathbb{N}_+$. The Sobolev space $W^{r,\infty}([0, 1]^d)$ is the set of functions belonging to $C^{r-1}([0, 1]^d)$ whose $(r - 1)$-th order derivatives are Lipschitz continuous with the norm

$$\|f\|_{W_\infty^r} := \max_{\alpha: |\alpha| \le r} \text{esssup}_{x \in [0,1]^d} |D^\alpha f(x)| < \infty.$$

We denote by $U^r([0, 1]^d)$ the unit ball of $W^{r,\infty}([0, 1]^d)$, i.e. $U^r([0, 1]^d) = \{f \in W^{r,\infty}([0, 1]^d) : \|f\|_{W_\infty^r} \le 1\}$. Note

$$\text{ess sup } f = \inf\{a \in \mathbb{R} : \mu(\{x : f(x) > a\}) = 0\}$$

where $\mu$ is Lebesgue measure.

### D.2 PROOF OF THEOREM 5

We follow the proof of theorem 1 in Yarotsky (2017). In our proof, we skip some details and focus on the constructions of ResNet. The details can be found in theorem 1 of Yarotsky (2017). Now let $f \in U^r([0, 1]^d)$ and $\boldsymbol{\alpha} \in \mathbb{N}^d$.

Let $N$ be a positive integer to be determined and $\mathbf{m} = (m_1, \ldots, m_d) \in \{0, 1, \ldots, N\}^d$. The function $\phi_{\mathbf{m}}$ is defined as the product

$$\phi_{\mathbf{m}}(\mathbf{x}) = \prod_{k=1}^d \psi \left( 3N \left( x_k - \frac{m_k}{N} \right) \right)$$

where

$$\psi(x) = \begin{cases} 1, & |x| < 1, \\ 0, & 2 < |x|, \\ 2 - |x|, & 1 \le |x| \le 2. \end{cases}$$

Let

$$f_1(\mathbf{x}) = \sum_{\mathbf{m} \in \{0,\ldots,N\}^d} \sum_{\boldsymbol{\alpha}: |\boldsymbol{\alpha}| < r} a_{\mathbf{m}, \boldsymbol{\alpha}} \phi_{\mathbf{m}}(\mathbf{x}) \left( \mathbf{x} - \frac{\mathbf{m}}{N} \right)^{\boldsymbol{\alpha}}$$

where $a_{\mathbf{m},\boldsymbol{\alpha}}$ are some specific coefficients when considering the locally Taylor expansion of $f$. Then by choosing

$$N = \left\lceil \left( \frac{r!}{2^d d r} \frac{\varepsilon}{2} \right)^{-1/r} \right\rceil \tag{27}$$

where $\lceil \cdot \rceil$ is the celling function, we have

$$\|f - f_1\|_\infty \le \frac{\varepsilon}{2}.$$

Now, we consider to approximate $\phi_{\mathbf{m}}(\mathbf{x})\left(\mathbf{x} - \frac{\mathbf{m}}{N}\right)^{\boldsymbol{\alpha}}$ by ResNet.

The following lemma follows directly from remark 17 that

**Lemma 18.** *Let $\mathbf{x} \in [0,1]^d$ and $g_1(x_1), \cdots, g_2(x_d) \in [-1,1]$. Then*

$$\left| \pi_n^d\left(g_1(x_1), g_2(x_2), \cdots, g_d(x_d)\right) - g_1(x_1)g_2(x_2)\cdots g_d(x_d) \right| \le ed4^{-n}, \quad n \ge 1 + \log_2 d$$

*for all $x_1, x_2, \cdots, x_d \in [-1,1]$.*

Moreover, we need the following lemma for the construction.

**Lemma 19.** *There is a ResNet $R(x) \in \mathcal{RN}\,(k=1, N=3, L=4)$ such that $R(x) = \psi(x)$ for any $x \in \mathbb{R}$.*

*Proof of lemma 19.* Lin & Jegelka (2018) has shown that the following operations are realizable by a single basic residual block of ResNet with one neuron: (a) Shifting by a constant: $R^+ = R + c$ for any $c \in \mathbb{R}$. (b) Min or Max with a constant: $R^+ = \min\{R, c\}$ or $R^+ = \max\{R, c\}$ for any $c \in \mathbb{R}$. (c) Min or Max with a linear transformation: $R^+ = \min\{R, \alpha R + \beta\}$ (or max) for any $\alpha, \beta \in \mathbb{R}$.

For the ResNet, $z^{[0]} = x$. In the first layer, we use one computational unit to compute $\mathrm{ReLU}(x+2)$ and two units to compute $x = (x)_+ - (-x)_+$. Then we output $z^{[1]} = \mathrm{ReLU}((x+2) - x + x = \mathrm{ReLU}(x+2)$ in the first layer. In the remaining layers, we only need one neuron per layer. we can output $z^{[2]} = z^{[1]} - 2(z^{[1]} - 2)$, $z^{[3]} = \min\{z^{[2]}, 1\}$ and $z^{[4]} = \max\{z^{[3]}, 0\} = \psi(x)$. $\qquad\square$

Then now we define

$$R_{\mathbf{m},\boldsymbol{\alpha}}(x) = \pi_n^{2d}\left( \psi(3Nx_1 - 3m_1), \cdots, \psi(3Nx_d - 3m_d), \pi_n^{\alpha_1}\left(x_1 - \frac{m_1}{N}\right), \cdots, \pi_n^{\alpha_d}\left(x_d - \frac{m_d}{N}\right) \right)$$

$$= \pi_n^{d+|\boldsymbol{\alpha}|}\left( \psi(3Nx_1 - 3m_1), \cdots, \psi(3Nx_d - 3m_d), \underbrace{x_1 - \frac{m_1}{N}, \cdots, x_1 - \frac{m_1}{N}}_{\alpha_1\,\text{times}}, \cdots, \underbrace{x_d - \frac{m_d}{N}, \cdots, x_d - \frac{m_d}{N}}_{\alpha_d\,\text{times}} \right).$$

By lemma 18 and note

$$|\psi(3Nx_i - 3m_i)| \le 1 \quad \text{and} \quad \left|x_i - \frac{m_i}{N}\right| \le 1 \quad \text{for} \quad i = 1, 2, \cdots, d$$

, we have

$$\left| R_{\mathbf{m},\boldsymbol{\alpha}}(\mathbf{x}) - \phi_{\mathbf{m}}(\mathbf{x})\left(\mathbf{x} - \frac{\mathbf{m}}{N}\right)^{\boldsymbol{\alpha}} \right| < e(r+d)4^{-n} = \varepsilon_0, \quad \mathbf{x} \in [0,1]^d, \tag{28}$$

by letting $n = \mathcal{O}(\log \frac{e(r+d)}{\varepsilon_0})$. Then with similar proof of Theorem 3 and lemma 19, we have a ResNet with $k = d+3$ to generate $R_{\mathbf{m},\boldsymbol{\alpha}}(\mathbf{x})$ such (28) satisfies while having $\mathcal{O}(d(r+d)\log e(r+d)/\varepsilon_0)$ weights. It follows from the proof of Theorem 4, we have a ResNet with $k = d+4$ can generate

$$\widetilde{f}(\mathbf{x}) = \sum_{\mathbf{m} \in \{0,\ldots,N\}^d} \sum_{\boldsymbol{\alpha}:|\boldsymbol{\alpha}|<r} a_{\mathbf{m},\boldsymbol{\alpha}} R_{\mathbf{m},\boldsymbol{\alpha}}(\mathbf{x})$$

while having

$$\mathcal{O}\left( (N+1)^d d^r d(d+r) \log \frac{e(r+d)}{\varepsilon_0} \right) \tag{29}$$

weights. From the proof of theorem 1 in Yarotsky (2017) we then have

$$|\widetilde{f}(\mathbf{x}) - f_1(\mathbf{x})| \leq 2^d d^r \varepsilon_0.$$

Let $\varepsilon_0 = \varepsilon/(2^{d+1}d^r)$. We have $\|\widetilde{f} - f_1\|_\infty \leq \varepsilon/2$. Thus,

$$\|f - \widetilde{f}\|_\infty \leq \|f - f_1\|_\infty + \|f_1 - \widetilde{f}\|_\infty \leq \varepsilon.$$

Now, substitute $\varepsilon_0 = \varepsilon/(2^{d+1}d^r)$ and $N$ with equation 27 into (29), the upper bound on the total weights of the ResNet are

$$\mathcal{O}\left(\left(\frac{r!}{2^d d^r}\frac{\varepsilon}{2}\right)^{-d/r} d^{r+1}(d+r)\log\frac{e(r+d)2^d d^r}{\varepsilon}\right) = \mathcal{O}_{d,r}\left(\varepsilon^{-\frac{d}{r}}\log\frac{1}{\varepsilon}\right).$$

By the well-known Stirling's approximation

$$\sqrt{2\pi r}\left(\frac{r}{e}\right)^r e^{\frac{1}{12r+1}} < r! < \sqrt{2\pi r}\left(\frac{r}{e}\right)^r e^{\frac{1}{12r}}, \tag{30}$$

the hidden constant $c(d,r)$ in the $\mathcal{O}_{d,r}$ notation can be bounded by

$$(\frac{2^{\frac{d+1}{r}}d}{r})^d < c(d,r) < (\frac{2^{\frac{d+1}{r}}d}{r})^d d^{r+2}(d+r)r \tag{31}$$

where $C$ is an absolute constant.

# E    PROOF OF THEOREM 6

In this appendix, we give the proof of Theorem 6. The proof of Theorem 6 is mainly based on the following lemma.

**Lemma 20.** *Fix the integer $d \geq 1$ and let $f : \mathbb{R}^d \to \mathbb{R}$ be a CPwL function. Then there exist affine functions $p_\alpha, q_\beta : \mathbb{R}^d \to \mathbb{R}$ such that $f$ can be written as the difference of positive convex functions:*

$$f = p - q, \quad where \quad p := \max_{1\leq\alpha\leq P} p_\alpha, \quad q := \max_{1\leq\beta\leq Q} q_\beta$$

*where $P, Q$ are some positive numbers.*

*Proof.* For any CPwL function $f : \mathbb{R}^d \to \mathbb{R}$, by theorem 1 in Wang & Sun (2005), there exists a finite set of affine linear functions $\ell_1, \ldots, \ell_k$ and a finite integer $M$ such that

$$f = \sum_{j=1}^{M} \sigma_j \left(\max_{i\in S_j} \ell_i\right)$$

where $S_j \subseteq \{1, \ldots, k\}$, $|S_j| \leq d+1$ and $\sigma_i \in \{+1, -1\}$ for all $i = 1, 2, \cdots, M$. We write

$$f = \sum_{j=1}^{M} \sigma_j \left(\max_{i\in S_j} \ell_i\right) = \sum_{j:\sigma_j=1} \max_{i\in S_j} \ell_i - \sum_{j:\sigma_j=-1} \max_{i\in S_j} \ell_i = p - q.$$

The last equation holds by the fact that the sum of convex functions is convex and the sum of CPwL functions is also CPwL. We can easily see $P, Q$ is bounded by $Md$. However, $M$ is an implicit number that may depend on the property of the CPwL function (e.g., the number of pieces, and the number of linear components). More details can be found in the proof details in Tarela & Martinez (1999); Wang & Sun (2005). □

Now we are ready for the proof of Theorem 6.

### E.1 Proof of Theorem 6

The proof is based on the observation

$$\max\{x, y\} = y + \text{ReLU}(x - y)$$

Now we construct a single-neuron per hidden layer ResNet with $k = d + 1$ to output $f$ exactly. We use the same notation as lemma 20. Let $z^{[0]} = \mathcal{A}_{d+1}(x) = (x, p_1(x))$. In the next block, we compute $\zeta^{[1]} = \mathcal{T}_1^{[1]}(z^{[0]}) = \text{ReLU}(p_2(x) - p_1(x))$ in the activation layer and $\gamma^{[1]} \mathcal{T}_2^{[1]}(z^{[0]}) = (0, \text{ReLU}(p_2(x) - p_1(x)))$ in the identity layer by choosing the appropriate weights. Then we can output

$$z^{[1]} = \gamma^{[1]} + z^{[0]} = (0, \text{ReLU}(p_2(x) - p_1(x)) + z^{[0]} = (x, \max\{p_1, p_2\}).$$

By repeatedly doing this, we can output $z^{[P]} = (x, \max_{1 \leq \alpha \leq P}\{p_\alpha\})$ in the $P$-th block. In the next two block, we output $z^{[P+2]} = (x, p(x) - q_1(x))$. Here we use two single-neuron blocks to compute $q_1(x) = (q_1)_+ - (-q_1)_+$. Then by the same operation, we can get the result $z^{[P+Q+2]} = (x, p - q)$ in the $(P + Q + 2)$-th block. Then the ResNet outputs $f = p - q$ exactly.

To approximate continuous functions, we give the following precise result.

**Corollary 21.** *For any continuous function $f : [0, 1]^d \to \mathbb{R}$, there is a ResNet*

$$R(x) \in \mathcal{RN}(d + 1, 1, L)$$

*with $L = \mathcal{O}_d(\omega_f(\varepsilon)^{-d})$ such that $\|f - R\|_{C([0,1]^d)} < \varepsilon$ where*

$$\omega_f(t) := \sup\{|f(x) - f(y)| : |x - y| \leq t\}.$$

The proof is directly from the proof of equation $(6), (7)$ of theorem 1 in Hanin (2019). One point of difference is that the continuous functions are assumed to be from $[0, 1]^d$ to $\mathbb{R}_+$ in that paper. Actually, the domain $[0, 1]^d$ of $f$ is compact so there exists a constant $C$ such that $f + C \geq 0$. Then our assumption can be converted into the same as theirs. Here we just briefly talk about their proof ideas.

They first sub-divide $[0, 1]^d$ into at most $\omega_f(\varepsilon)^{-d}$ cubes of side length at most $\omega_f(\varepsilon)$. Then, they subdivide each such smaller cube into $d$ ! copies of the standard simplex $\{\mathcal{P}_j\}_j$ (which has volume $1/d$ !) rescaled to have side length $\omega_f(\varepsilon)$. Then define a CPwL function $f_\varepsilon$ which equals to $f$ on the vertices of $\mathcal{P}_|$ and is affine on their interiors. Then

$$\|f - f_\varepsilon\|_{C([0,1]^d)} \leq \varepsilon.$$

Then by Lemma 20 and Theorem 6. We can use ResNet belonging to $\mathcal{RN}(d + 1, 1, L)$ to generate $f_\varepsilon$ where $L = \mathcal{O}(\omega_f(\varepsilon)^{-d})$.

### E.2 Some discussions

Piecewise linear interpolation holds a significant position in approximation theory, as it is a basic method of approximating functions. Therefore, studying the expressive power of neural networks for piecewise linear functions becomes particularly important. Every ReLU FNN is a CPwL function, bringing forth intriguing questions about the relationship between ReLU neural networks and CPwL functions, such as the network size required to represent an arbitrary CPwL function. According to Arora et al. (2016), any CPwL function from $\mathbb{R}^d$ to $\mathbb{R}$ can be represented by a deep ReLU network with depth at most $\lceil \log_2(d + 1) \rceil + 1$ and sufficient width. Later, Hanin (2019) and DeVore et al. (2021) revealed that a ReLU network of fixed width ($d + 3$ and $d + 2$ respectively) can generate any CPwL function with sufficiently large depth. In our new result for ResNet, Theorem 6 shows that a ResNet with one neuron per activation layer can represent any CPwL function over $\mathbb{R}^d$ given enough depth. While Lin & Jegelka (2018) demonstrated the construction of ResNet for approximating step constant functions, its methods are limited to generating CPwL functions. Our Proposition 1 aligns with DeVore et al. (2021) and shows that a sufficiently deep ReLU network with width $d + 2$ can generate any CPwL function over $[0, 1]^d$. However, ResNet, with the same number of neurons, offers fewer tunable parameters, and superior training efficiency for very deep networks due to its ability to solve the issue of gradient exploding/vanishing.

## F PROOF OF THEOREM 8

Before proceeding to the proof of Thm. 8, we first discuss more explanations and a conclusion.

To approximate function $f(x_1, \cdots, x_d) = \sum_{q=0}^{2d} g\left(\sum_{i=1}^{d} \lambda_i \phi_q(x_i)\right) \in K_C$ i.e., $g$ and $\phi_q (q = 1, 2, \cdots, d)$ are Lipschitz continuous functions, one can use piecewise linear splines to approximate $g$ and $\phi_q$. Concretely,

$$\widetilde{g}(x) = \sum_{k=1}^{n} \omega_k \sigma(x - y_k) \approx g(x) \quad \text{and} \quad \widetilde{\phi_{q,i}}(x_i) = \sum_{j=1}^{n} c_{qj,i} \sigma(x_i - y_{qj,i}) \approx \lambda_i \phi_q(x_i)$$

for any $x \in [0, 1]$ where $w_k, c_{qj} \in \mathbb{R}, y_k \in [0, 1], y_{qj} \in [0, 1]$. Then we can use

$$f(x_1, \cdots, x_d) = \sum_{q=0}^{2d} \widetilde{g}\left(\sum_{i=1}^{d} \widetilde{\phi_{q,i}}(x_i)\right) \in K_{n,n}$$

to approximate $f$ and can achieve the approximation rate in Theorem 22.

**Theorem 22** (Theorem 4, Lai & Shen (2021)). *We have*

$$\sup_{f \in K_C} \inf_{s \in \mathcal{K}_{n,n}} \|f - s\|_{C([0,1]^d)} \leq \frac{C(2d+1)^2}{n}$$

*where*

$$\mathcal{K}_{n,n} = \left\{ \sum_{q=0}^{2d} \sum_{k=1}^{n} w_k \sigma\left(\sum_{i=1}^{d} \sum_{j=1}^{n} c_{qj} \sigma(x_i - y_{qj}) - y_k\right), w_k, c_{qj} \in \mathbb{R}, y_k \in [0, 1], y_{qj} \in [0, 1] \right\}.$$

We then ready to prove Thm. 8.

### F.1 PROOF OF THEOREM 8

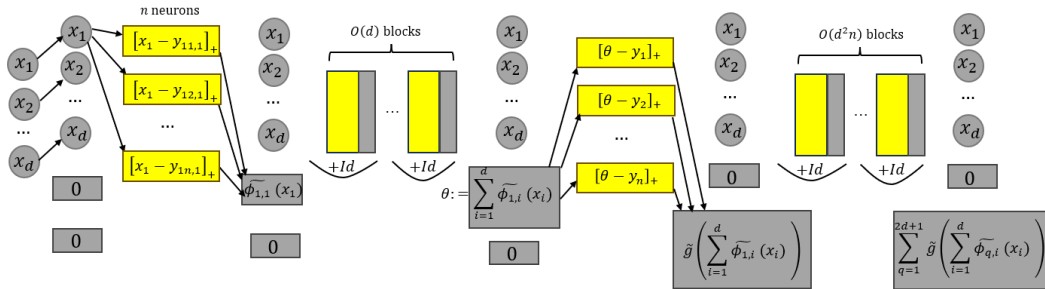

Figure 9: Illustration for generating the KST structure in $\mathcal{K}_{n,n}$. Grey represents the identity layer and yellow represents the activation layer.

Based on the above discussion, it suffices to show ResNet $R(x) \in \mathcal{RN}(d+2, n, L)$ with $L = \mathcal{O}(d^2)$ can generate any function in $\mathcal{K}_{n,n}$ using $\mathcal{O}(d^2 n)$ non-zero weights. We then get the conclusion by Theorem 22.

Assume $x \in [0, 1]^d$. For a ResNet, let $z^{[0]} = (x, 0, 0)$. Then we can output

$$\zeta^{[1]} = \mathcal{T}_1^{[1]}(z^{[0]}) = (\sigma(x_1 - y_{11,1}), \sigma(x_1 - y_{12,1}), \cdots, \sigma(x_1 - y_{1n,1})).$$

in the next layer. By choosing some weights, we can compute

$$\gamma^{[1]} = V_1 \zeta^{[1]} = \begin{bmatrix} 0 & 0 & \cdots & 0 \\ 0 & 0 & \cdots & 0 \\ \vdots & \vdots & \ddots & \vdots \\ 0 & 0 & \cdots & 0 \\ c_{11,1} & c_{12,1} & \cdots & c_{1n,1} \\ 0 & 0 & \cdots & 0 \end{bmatrix}_{d+2,n} \cdot \zeta^{[1]} = (0_d, \widetilde{\phi_{1,1}}(x_1), 0).$$

Thus, the output of the first block is $z^{[1]} = z^{[0]} + \gamma^{[1]} = (x, \widetilde{\phi_{1,1}}(x_1), 0)$. By repeatedly doing this in the next $d$ blocks, we can generate

$$z^{[d]} = (x, \sum_{i=1}^{d} \widetilde{\phi_{1,i}}, 0).$$

In the next layer, we compute

$$\zeta^{[d+1]} = \mathcal{T}_1^{[d+1]}(z^{[d]}) = (\sigma(z_{d+1}^{[d]} - y_1), \sigma(z_{d+1}^{[d]} - y_2), \cdots, \sigma(z_{d+1}^{[d]} - y_n)).$$

Then by choosing some appropriate weights, we have

$$z^{[d+1]} = \left( x, 0, \widetilde{g} \left( \sum_{i=1}^{d} \widetilde{\phi_{1,i}}(x_i) \right) \right).$$

Also by repeatedly doing the operation, we can get in the $(d^2 + 2d + 1)$-th block that

$$z^{[(d+1)^2]} = \left( x, -, \sum_{q=0}^{2d} \widetilde{g} \left( \sum_{i=1}^{d} \widetilde{\phi_{q,i}}(x_i) \right) \right).$$

See Figure 9 for an illustration.

Moreover, in each block, only $\mathcal{O}(n)$ weights are non-zero. Thus, the total tunable weights of this neural network is $\mathcal{O}(d^2 n)$

## G    Supplementary discussions

### G.1    Related work

These are supplementary details to the related work section (Sec. 1.1) of this paper.

**Universality.** The universality of a function family implies that this family is dense in the space of continuous functions, meaning it can approximate any continuous function with arbitrary precision. In the earlier years, Cybenko (1989) made a groundbreaking argument by demonstrating that shallow neural networks equipped with sigmoid activation functions possess universal approximation properties. This activation condition was further expanded by Pinkus (1999) who showed that shallow networks employing non-polynomial activation functions also exhibit universal approximation capabilities. In recent years, the universality of narrow deep networks has also attracted considerable attention. Hanin & Sellke (2017) determined that a deep ReLU neural network must have a minimum width of $d + 1$ to ensure universality, where $d$ is the input dimension. Kidger & Lyons (2020) then showed that deep narrow networks with any continuous activation function can achieve universality, provided that the activation satisfies a very mild condition. Over the past decades, a variety of network architectures have been developed to cater to diverse tasks and objectives, extending beyond feedforward ReLU networks. The universal approximation theorem has been proven for multiple network architectures, including: standard deep ReLU CNN Zhou (2018; 2020), deep ReLU CNNs with classical structures He et al. (2022), continuous-time recurrent neural network (RNN) Li et al. (2020; 2022b), continuous-time ResNet Li et al. (2022a), ResNet Lin & Jegelka (2018), and ResNet for finite-sample classification tasks Hardt & Ma (2016).

**Approximation Capabilities.** There has been substantial progress in enhancing our theoretical understanding of neural networks. Some studies have focused on comparing the expressive power of both shallow and deep neural networks, examining their respective capabilities (e.g., Arora et al. (2016); Eldan & Shamir (2016); Liang & Srikant (2016); Telgarsky (2016); Yarotsky (2017); Poggio et al. (2017)). Some others have quantified the number of linear regions within deep neural networks, casting light on their complexity (e.g., Montufar et al. (2014); Serra et al. (2018); Arora et al. (2016)). Moreover, constructive methods have been utilized to probe the approximation capabilities across different function classes. Notably, researchers have delved into the optimal approximation of continuous functions (e.g., Shen et al. (2022b); Yarotsky (2018)), the optimal approximation of smooth functions (e.g., Yarotsky (2017); Lu et al. (2021); Montanelli & Du (2019)), and the approximation of analytic functions (e.g., Wang et al. (2018); Schwab & Zech (2021)). These diverse

investigations collectively deepen our understanding of both the potential and constraints of neural networks in approximating various types of functions.

**Perspectives on the Curse of Dimensionality.** The 'curse of dimensionality' coined by Bellman (1957) refers to a phenomenon that a model class will suffer an exponential increase in its complexity as the input dimension increases. This impact is explicitly observed in ReLU networks, as well-documented in Yarotsky (2017). Importantly, the curse of dimensionality, not limited to MLPs, is also a challenge for almost all classes of function approximators aiming to uniformly approximate in the Lipschitz domain (have sufficient regular boundary) on some compact subset of a metric space due to the entropy limitation Kolmogorov & Tikhomirov (1959).

More specifically, any continuous function approximator[6] will suffer the curse of dimension in the smooth function space $C^r$ DeVore et al. (1989) because the metric entropy of the unit ball in $C^r$ with respect to the uniform topology is $\Theta(\varepsilon^{-d/r})$. The property is applied to ReLU neural networks in Thm. 3 Yarotsky (2017). In an attempt to mitigate the curse of dimensionality, initial strategies involved the consideration of specialized function spaces whose metric entropy is expected to reduce such as analytical functions Wang et al. (2018), bandlimited functions Montanelli et al. (2019), Korobove space Montanelli & Du (2019). Specifically, for analytical functions on $[-1+\delta, 1-\delta]^d$ (Wang et al. (2018)), ReLU networks can achieve the exponential approximation rate of $\mathcal{O}(\exp\{-d\delta(e^{-1}L^{1/2d}-1)\})$ for any small $\delta > 0$. For band-limited functions, Montanelli et al. (2019) shows that ReLU networks with a depth of $\mathcal{O}_f(\log^2 \frac{1}{\varepsilon})^7$ and a width of $\mathcal{O}_f(\frac{1}{\varepsilon^2}\log^2 \frac{1}{\varepsilon})$ can achieve an $\varepsilon$-approximation. Additionally, Montanelli & Du (2019) demonstrates that ReLU networks with a total neuron count of $\mathcal{O}(\varepsilon^{-1/2}(\log 1/\varepsilon)^{\frac{3}{2}(d-1)+1}\log d)$ can approximate any function in Korobov space using a sparse grids structure.

More recently, researchers have shifted their focus toward the structure of neural networks, suggesting a potential solution to circumvent the curse of dimensionality. Some researchers consider the parameters-sharing method (e.g., repeated-composition structure Zhang et al. (2023), CNN Zhou (2018)). A more recent trend aims to serve neural networks as discontinuous function approximators, thereby examining neural networks with novel activation functions (e.g., ReLU-floor activation Shen et al. (2020), ReLU-sine-exponential activation Jiao et al. (2023), floor-exponential-step activation Shen et al. (2021), activation composed of triangular-wave and softsign function Shen et al. (2022a)). Meanwhile, they incorporate the Kolmogorov Superposition Theorem (KST)(see Sec. 5) to tackle the curse of dimensionality, yielding promising theoretical results. For instance, Yarotsky (2021) and Shen et al. (2022a) show that deep networks with particular activation can approximate any continuous functions over $[0,1]^d$ with complexity $O(d^2)$. Moreover, Lai & Shen (2021) and He (2023) show that ReLU networks can overcome the curse of dimensionality within a special function class derived from KST. However, the failure of these model classes in practice is due to the discontinuity of the function approximators, wherein even minor perturbations in the training data can lead to chaotic changes in the input-output relationship. Consequently, to circumvent the curse of dimensionality, it is imperative to make appropriate choices within the unstable model class and the restricted objective function space.

### G.2 RELATIONS WITH DYNAMIC SYSTEMS

ResNet shares a deep correlation with dynamic systems, to the point where one could consider ResNet as a form of discrete dynamic system. In this subsection, we will briefly discuss the relationship between ResNet and dynamic systems.

Dynamic systems is generally described by an ordinary differential equation (ODE):

$$\frac{d}{dt}z(t) = f_{\theta(t)}(z(t)), \quad \theta(t) \in \Theta, \quad t \in [0, T], \quad z(0) = x \in \mathbb{R}^n.$$

---

[6] In the context, we aim to approximate all functions in a space $\mathcal{F}$ using a model class as an approximator (e.g., neural networks). We achieve this by choosing different parameters for different functions, meaning the parameters $\theta \in \Theta$ can be seen as a mapping of the target functions, i.e., $\theta = h(f)$ where $h : \mathcal{F} \to \Theta$. If this mapping $h$ is continuous, we refer to the approximator as a continuous approximator.

[7] The subscript $f$ implies the hidden constant depends on $f$ including the domain and the dimension. More detail can be found in the paper.

where $f_{\theta(t)} : \mathbb{R}^n \to \mathbb{R}^n$ is the dynamics function of this dynamical system, and $\Theta$ is the parameter space. $z(T)$ can be regarded as a function of $x$, denoted by $\varphi_\theta(x)$ which is known as the flow map. Then we can use $\mathcal{L} \circ \varphi_\theta(x) \circ \mathcal{A}$ to approximate a target function $f : \mathbb{R}^d \to \mathbb{R}$ where $\mathcal{L}, \mathcal{A}$ are affine transformations.

By using the Euler method to discretize the ODE, we have $z_{s+1} = z_s + \delta \cdot f_{\theta_s}(z_s)$ for some small $\delta > 0$ for $s = 1, 2, \cdots, S-1$ where $T = \delta S$. In our results, $f_{\theta_s}(z_s)$ can be realized by a shallow constant-neuron ReLU network block, in which the number of tunable parameters can be absolute constant which is independent of $d$. Then we can have $z_S \approx z(T) = \varphi_\theta(x)$.

Thus, our results reveal that a continuous-depth network generated via a dynamical system possesses significant approximation capabilities even if its dynamics function is dimension-independent and realized by a shallow constant-neuron ReLU network block. Moreover, there is a comprehensive review of the correlations between deep learning and dynamic systems in the recent work Zhang et al. (2023).

## H  EXPERIMENTS

In this appendix, we specify the experiment setting in Section 6. First, we assume that an appropriate algorithm can effectively manage the optimization error (e.g., Adam optimizer (Kingma & Ba, 2014)). We then choose a sufficiently complex target function to ensure that the approximation error is the dominant factor. Specifically, we utilize the following function to test the universal approximation capability of b-ResNet.

$$f(x) = \sum_{i=1}^{m} [a_i \prod_{j \in S_i^1} x_j + b_i \sin(\prod_{k \in S_i^2} x_k)], \tag{32}$$

where $x \in [-1, 1]^d$, $S_i^1$ and $S_j^1$ are the index sets randomly sampled from $\{1, ..., d\}$ with replacement, $a_i$ and $b_i$ are constant coefficients, and $m$ is the total number of terms. Specifically, we set the parameters for the defined function as $d \in \{100, 200, 300\}$, $m = d/10$, $\text{card}(S_i^1) \leq \sqrt{d}$, $\text{card}(S_i^2) \leq \sqrt{d}$, $a_i = 1$ and $b_i = 0.1$. We then compare b-ResNet with fully connected NN for approximating the defined function, with network structure as $\mathcal{RN}(d+1, n, d/10)$ for $n \in \{10, 20, 30\}$, and $\mathcal{NN}(d+1, d/10)$, respectively. Next, We conduct Quasi Monte Carlo sampling over $[-1, 1]^d$ with $1000 \cdot d$ samples and use 90% for training and 10% for testing. We optimize the network parameters using Adam (Kingma & Ba, 2014) with a learning rate of $1e^{-3}$ and present the test performance over iteration.

The results include the mean square error (MSE) and the infinite norm error (MAX) on testing samples. Further, we also compared the testing performance under different training losses, including MSE and MAX, are shown in the following figures.

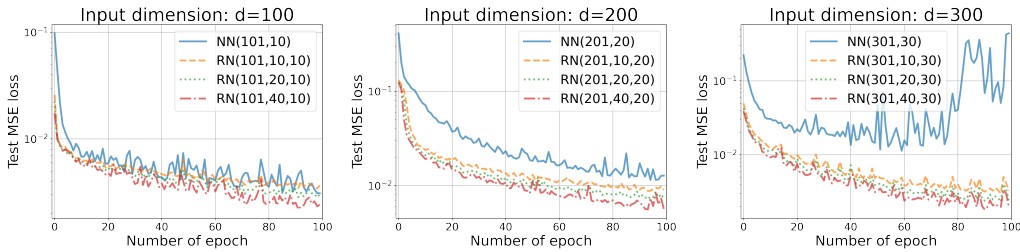

Figure 10: Comparison of testing MSE loss by training with MSE loss.

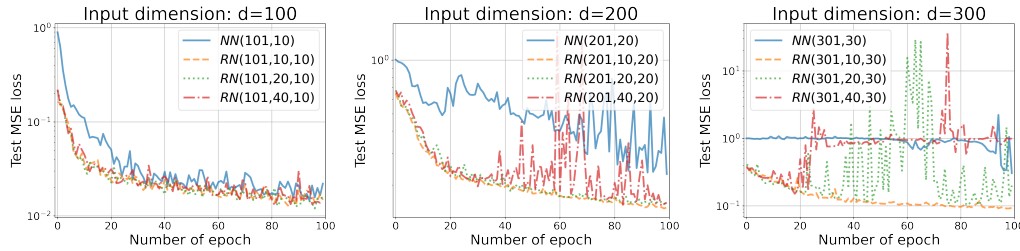

Figure 11: Comparison of testing MSE loss by training with MAX loss.

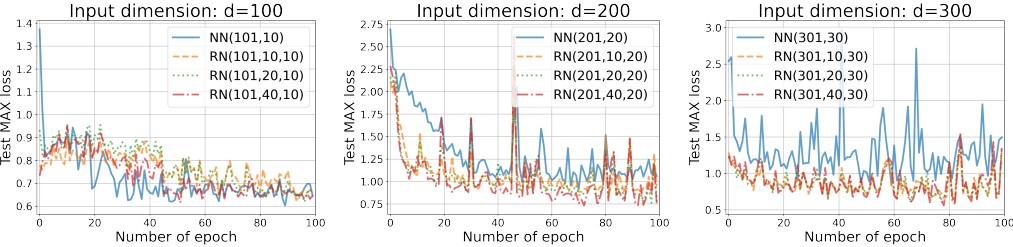

Figure 12: Comparison of testing MAX loss by training with MSE loss.

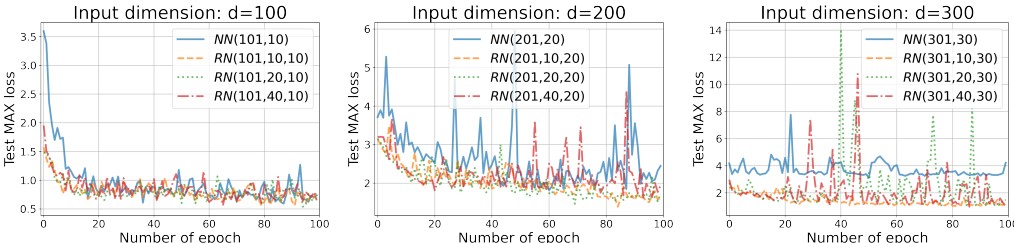

Figure 13: Comparison of testing MAX loss by training with MAX loss.

