# OpenReview forum: "Characterizing ResNet's Universal Approximation Capability"
_ICLR.cc/2024/Conference — Submitted to ICLR 2024_

### Official Review · Reviewer_mYBU · 2023-10-28

**Soundness:** 4 excellent
**Presentation:** 3 good
**Contribution:** 3 good
**Rating:** 8
**Confidence:** 5

**Summary:**

The authors clearly and correctly derive quantitative universal approximation guarnatees for the ResNet architecture.  Furthermore, they exhibit a class which can be efficiently approximated (i.e. without the curse of dimensionality) using the ResNet model.

**Strengths:**

The results are very interesting, definitely publishable, and I will likely be using/quoting them myself in the near future ;).  Thanks for the very nice contribution :)

Personally: I was looking for such a result in the literature, so I must say I'm happy its added.  Good work!

**Weaknesses:**

**1) Lack of Explanation in the "Perspectives on the Curse of Dimensionality" paragraph need clarification/explanation/motivation.**

The way this section is written leaves the reader, not familiar with constructive approximation in the dark, thinking some "magic is happening" where "there be dragons".  Worry not!, here is how to fix this issue:

1. the curse of dimensionality is not only something that MLPs experience, but every (reasonable) class of function approximators which are uniformly approximating in the Lipschitz ball on some compact subset of a metric space.  Indeed, this is due to metric entropy limitations and the result is essentially due to Kolmogorov [1].  I think the authors should add 1 sentence at the end of page 2 to explain how this is a **generic** problem, and that the reader should expect this to **always** be the case.

2. By the same token as (1), above, the reason which people consider speacialized classes of functions as a viable way to circumvent the curse of dimensionality is again rooted in metric entropy number arguments.  Indeed, [1] or even [2] show that the metric entropy (which is a non-linear notion of comprehensibility of a function class), or even linear notions such as Kolmogorov linear widths, are not cursed by dimensionality for smooth classes.  Indeed, the metric entropy of the unit ball in $C^k$ wrt the uniform topology is $\Theta(\varepsilon^{-r/d})$ so one can expect that the optimal (non-linear) approximator can achieve such rates.  These ideas should be communicated to the reader atop page $3$ so that it does not seem that consider "special functions classes" is a "magic trick" :)

Fyi, the relationship between linear and non-linear measures of compressibility (the above which I mention) are given by Carl in their wonderful paper [3].

3. There similar reason justified why the MLPs with super expressive activation in the following paragraph.  Namely, due to the (near) discontinuity of these classes, as a function of their parameters.  See [4].  So again, there is a simple, conceptual explanation for why these approaches work, but also why they should not be employed (since beating the curse of dimensionality in this way is only possibly by models whose input-output relation changes chaotically if the training data is perturbed only a tiny bit, since minor parameter changes result in dramatically different functions.  For instance, this is very clear in Theorem 1 of [5].

I strongly feel that, points 1-3, should be communicated to the reader.  Albeit more conclusively due to the page limitations, if one even considers embarking on a meaningful discussion on the curse of dimensionality, and why/how it can be beaten.  More critically, the authors should, in such a discussion, highlight the reproductions of beating it; namely: the choice between an unstable model class (in its parameters) or a restriction of the functions which can be effectively approximated.


-----

**2) Suboptimal Rates Due to Polynomial Approximation and Not Quantization**

The proof strategy is akin to [6] (or its quantitative version in Proposition 53 - [7]) where one first approximates polynomials using the network class (here ResNet and in those papers deep but narrow MLPs), then one concludes by relating the polynomials to the target function being approximated (e.g. via quantitative Stone-Weirestrass theorems using Bernstein polynomials).  However, due to the "triangle inequality"-based argument these rates are suboptimal compared to the more modern quantization-based arguments of [8] which are possible for structures with recurrence/depth like MLPs ResNets.  For this reason, the rates which the authors derive are suboptimal by an exponential factor of $1/2$.

For example, in Theorem 3.3 [9], the author exhibits the existence of an MLP with ReLU activation achieving the optimal (with the optimality given, for example, by Theorem 3.1 [9] ) depth of $\mathcal{O}(\varepsilon^{-r/(2d)}$ is required to approximate any function in the unit ball of $H^r$ (as opposed to the current manscript's ResNet guarantee of $\tilde{\mathcal{O}}(\varepsilon^{-r/d})$.

** 3) Minor Comment(s)**

Some figures, e.g. Figure 6 in the appendix, seem not be expored as pdf or eps files making them pixelated.  Please fix this.


-- References --

[1] Kolmogorov, A. N., and V. M. Tikhomirov. "ϵ-entropy and ϵ-capacity." Uspekhi Mat. Nauk 14 (1959): 3-86.

[2] Birman, Mikhail Shlemovich, and Mikhail Zakharovich Solomyak. "Piecewise-polynomial approximations of functions of the classes W_p^α." Matematicheskii Sbornik 115, no. 3 (1967): 331-355.

[3] Carl, Bernd. "Entropy numbers, s-numbers, and eigenvalue problems." Journal of Functional Analysis 41, no. 3 (1981): 290-306.

[4] DeVore, Ronald A., Ralph Howard, and Charles Micchelli. "Optimal nonlinear approximation." Manuscripta mathematica 63 (1989): 469-478.

[5]  Shen, Zuowei, Haizhao Yang, and Shijun Zhang. "Deep network approximation: Achieving arbitrary accuracy with fixed number of neurons." The Journal of Machine Learning Research 23, no. 1 (2022): 12653-12712.

[6] Kidger, Patrick, and Terry Lyons. "Universal approximation with deep narrow networks." In Conference on learning theory, pp. 2306-2327. PMLR, 2020.

[7] Kratsios, Anastasis, and Léonie Papon. "Universal approximation theorems for differentiable geometric deep learning." The Journal of Machine Learning Research 23, no. 1 (2022): 8896-8968.

[8] Shen, Zuowei, Haizhao Yang, and Shijun Zhang. "Optimal approximation rate of ReLU networks in terms of width and depth." Journal de Mathématiques Pures et Appliquées 157 (2022): 101-135.

[9] Yarotsky, Dmitry, and Anton Zhevnerchuk. "The phase diagram of approximation rates for deep neural networks." Advances in neural information processing systems 33 (2020): 13005-13015.

**Questions:**

1. Can you please incorporate the discussion I added above, in the weaknesses.  I think the explanation of the curse of dimensionality is not satisfactory nor insightful, but it can be made 100% by adding the above points.

2. Why not deploy the quantization argument of [8] instead of approximating polynomials as in [6] (or its quantitative version in Proposition 53 of [7])?

That said, changing this is not feasible/worth it at this point, so I think its appropriate if the authors comment on this point.  Since, it is at the core of providing rates in their main results.


-- Reused references as Weaknesses section - In respective order --

[6] Kidger, Patrick, and Terry Lyons. "Universal approximation with deep narrow networks." In Conference on learning theory, pp. 2306-2327. PMLR, 2020.

[7] Kratsios, Anastasis, and Léonie Papon. "Universal approximation theorems for differentiable geometric deep learning." The Journal of Machine Learning Research 23, no. 1 (2022): 8896-8968.

[8] Shen, Zuowei, Haizhao Yang, and Shijun Zhang. "Optimal approximation rate of ReLU networks in terms of width and depth." Journal de Mathématiques Pures et Appliquées 157 (2022): 101-135.

---

> ### Author Response · Authors · 2023-11-18
> **Reply from authors**
>
> # Response to reviewer's comments and questions
>
> Dear reviewer,
>
> Thank you for your thorough reading of our manuscript and for your constructive feedback. We would like to express our sincere appreciation for the very careful and instructive comments! We have carefully prepared a revised manuscript and uploaded it for reference, taking your suggestions and those from other reviewers into account. For your questions, we response to them one by one as follows.
>
> **Q1:Can you please incorporate the discussion I added above, in the weaknesses. I think the explanation of the curse of dimensionality is not satisfactory nor insightful, but it can be made 100\% by adding the above points.**
>
> Response to Q1: We agree with your very helpful suggestions, and we have thoroughly implemented them in the revised manuscript  (see **Sec. 1.1, pages 2-3**).
>
> Specifically, we have made several modifications according to your suggestions to **the part 'Perspective on the curse of dimensionality'** in **Sec. 1.1, on pages 2-3**, to clarify and improve our discussion.
>
> **Response to comment 1.1**: *...I think the authors should add 1 sentence at the end of page 2 to explain how this is a generic problem, and that the reader should expect this to always be the case.*
>
> We have added a sentence **at the end of the first paragraph** to tell readers that the curse of dimensionality is a pervasive challenge in function approximation, not exclusive to MLPs, resulting from metric entropy restrictions.
>
> **Response to comment 1.2**: *...These ideas should be communicated to the reader atop page 3 so that it does not seem that consider "special functions classes" is a "magic trick"*
>
> We have expanded our discussion **at the beginning of the second paragraph** to explain why we consider specialized function classes as a strategy to mitigate the curse of dimensionality. This rationale is grounded on metric entropy number arguments, and we appreciate your suggestion to highlight this.
>
> **Response to comment 1.3**:*There similar reason justified why the MLPs with super expressive activation in the following paragraph. Namely, due to the (near) discontinuity of these classes, as a function of their parameters....*
>
> Addressing your comment on the role of super-expressive activation functions, we have added a sentence **in the third paragraph.** We emphasize that these activation functions can potentially aid neural networks to overcome the curse of dimensionality due to their ability to become discontinuous function approximators. Finally, **at the end of the third paragraph**, we have pointed out that the choice between unstable model class approximators and constrained target function classes is a necessary balance to avoid the curse of dimensionality.
>
> We believe these revisions substantially improve the paper quality. Thank you very much again!
>
> **Comment 2**: *Suboptimal Rates Due to Polynomial Approximation and Not Quantization.*; and
>
> **Q2: Why not deploy the quantization argument of [8] instead of approximating polynomials as in [6] (or its quantitative version in Proposition 53 of [7])?**
>
> Response to Comment 2 and Q2: We appreciate your comments regarding the suboptimal rates in our work, particularly in comparison to modern quantization-based arguments. We acknowledge the limitations of the approach adopted in our paper and would like to clarify our initial selection of polynomial approximation was motivated by its capacity to construct a b-ResNet (ResNet with constant width) that achieves the approximation rate. In our view, this represents an intriguing result with potential independent interest. Furthermore, we can utilize this result to demonstrate that the upper bounds will experience a reduction by a factor of 'd' (as per Theorem 3).
>
> However, we concur that pursuing the optimal approximation rate using ResNet represents a compelling research direction. Given the quantization-based arguments, we hypothesize that ResNets might also attain the optimal order, a topic we plan to explore in future work. We have addressed this point in our revised manuscript, specifically in the final paragraph of **Section 4.3, Page 8**, acknowledging the potential for improved rates with alternative methodologies.
>
> On a separate note, in response to your minor comments on the issues related to the figures, we have address the concerns for some figures (see e.g., Fig. 6 in the updated supplementary materials). As it is a bit time consuming to, we decided to update those we have done first and continue to refine the remaining figures. We will be able to improve all the figures in the final version.
>
> Again, thank you for your valuable feedback.

---

> > ### Comment · Reviewer_mYBU · 2023-11-22
> > **Thanks**
> >
> > Dear Authors,
> >
> > Thank you very much for incorporating the feedback, and for considering my suggestion for optimal rates as future research.  I think, with these modifications, the paper is in nice shape :)

---

> > > ### Author Response · Authors · 2023-11-23
> > > **Thanks for Reviewer Feedback and Appreciation for Constructive Review Process**
> > >
> > > Dear Reviewer,
> > >
> > > Thank you very much for your positive feedback and for your thoughtful suggestions during the review process. We are pleased to hear that you find the revised paper in good shape and your constructive feedback and insightful comments have greatly helped improve our work. Once again, thank you for your time and valuable input.
> > >
> > > Best regards

---

> ### Author Response · Authors · 2023-11-22
> **Gentle reminder**
>
> Dear reviewer,
>
> We hope this message finds you well. Thanks for your instructive comments again. We are writing to kindly remind you that the rebuttal stage deadline is due in a day. We fully understand the  busy schedule you may have, and we sincerely appreciate your time and effort put into this process. We are eager to learn if we have adequately addressed your concerns and whether you have any further questions. Your feedback is instrumental and valuable in enhancing the quality of our paper. Thanks for your time and effort again!

---

### Official Review · Reviewer_Avff · 2023-10-28

**Soundness:** 3 good
**Presentation:** 4 excellent
**Contribution:** 1 poor
**Rating:** 3
**Confidence:** 4

**Summary:**

This paper aims to extend e Lin & Jegelka (2018) which shows that ResNet with single-neuron can approximate any step function to approximate any CPwL function. Thus ResNet can become an optimal approximator.

Reviewer's concern is mostly the relationship between the submission with [1] (seems not cited and discussed in the paper). All the two papers are based on spline approximation and result in an O(1) channel ResNet. From the reviewer's perspective, it's essential to discuss different dependencies of dimension and differences in the structure before the paper is accepted.

The reviewer is open to increasing the score if this problem can be resolved during the interactive review process of ICLR.

[1] Oono, Kenta, and Taiji Suzuki. "Approximation and non-parametric estimation of ResNet-type convolutional neural networks." International conference on machine learning. PMLR, 2019.

**Strengths:**

Solid and well-written paper, construct a practical resnet with optimal apprxoimaton ability.

**Weaknesses:**

See above. missing literature and the comparison to the previous work.

**Questions:**

See above

---

> ### Author Response · Authors · 2023-11-18
> **Reply from authors**
>
> Dear reviewer,
>
> Thanks for your effort to review our paper. *Reviewer's concern is mostly the relationship between the submission with [1] (seems not cited and discussed in the paper). Both papers are based on spline approximation and result in an O(1) channel ResNet. From the reviewer's perspective, it's crucial to discuss different dependencies of dimension and differences in the structure before the paper is accepted.*
>
> We appreciate your insightful comments and for drawing our attention to the work in [1], which sets bounds on the errors of ResNet-type CNNs when approximating a class of smooth functions. We concur that a comparative analysis with the approximation capabilities of ResNet-type CNNs would indeed strengthen our paper. As such, we have extended our discussion to the work [1] in our revised manuscript. Please refer to the revised draft, specifically the last part in **Section 1.1, Page 3.**
>
> Here, we provide a brief summary of the revisions:
>
> Both our study and [1] delve into the approximation capabilities of ResNet-type neural networks. However, several key differences in the settings, results, and methodologies are worth noting.
>
> First, the settings and neural network architectures are different. Our work zeroes in on Fully Connected Neural Networks (FNNs) in ResNet, while [1] investigates Convolutional Neural Networks (CNNs) in ResNet. FNN and CNN possess distinct network structures, and our analysis, just like that of [1], leverages the unique structures of FNN and CNN, respectively. Consequently, results derived from one do not directly translate to the other.
>
> Second, the results are different. Our paper offers a novel construction of a simplified ResNet FNN structure, known as b-ResNet, and comprehensively delineates both the lower and upper bounds of its function approximation capability. Notably, we demonstrate that it achieves comparable approximation accuracy to ReLU FNN but with a significantly reduced number of tunable weights (a reduction by a factor of d, where d is the input dimension). In contrast, [1] establishes that a ResNet CNN can attain the same approximation accuracy as block-sparse FNNs, albeit with an order-wise equivalent number of tunable weights. However, it remains uncertain from [1] whether a ResNet CNN would require fewer weights. Furthermore, our analysis also reveals that ResNet FNN can approximate a class of smooth functions, leveraging the Kolmogorov Superposition Theorem, with the number of tunable weights only being polynomial in the input dimension d, thereby implying no curse of dimensionality. Such results are not reported in [1].
>
> Third, the approaches we adopted diverge significantly, despite both [1] and our work employing the standard spline approximation to establish some of the results. In our study, we demonstrate that the b-ResNet FNN can be implemented by ReLU FNN, and we utilize this relationship to set the *lower* bounds on the approximation capability of b-ResNet FNN. We establish the upper bounds by directly constructing ResNet FNN to approximate different function classes. In contrast, the authors in [1] illustrate that any block-sparse FNN can be realized by ResNet CNN, and they leverage this relationship to set the *upper* bounds of the approximation capability of ResNet CNN.
>
> In conclusion, we believe that this added comparison will offer a more holistic view of the ResNet approximation capability landscape. We are grateful for your valuable feedback and look forward to incorporating it into an improved version of our paper.
>
> References:
>
> [1] Oono, Kenta, and Taiji Suzuki. "Approximation and non-parametric estimation of ResNet-type convolutional neural networks." International conference on machine learning. PMLR, 2019.

---

> ### Author Response · Authors · 2023-11-22
> **Gentle reminder**
>
> Dear reviewer,
>
> We hope this message finds you well. We are writing to kindly remind you that the rebuttal stage deadline is due in a day. We fully understand the  busy schedule you may have, and we sincerely appreciate your time and effort put into this process. We are eager to learn if we have adequately addressed your concerns and whether you have any further questions. Your feedback is instrumental in enhancing the quality of our paper. Thanks for your time and effort again!

---

> ### Author Response · Authors · 2023-11-23
> **Gentle Reminder: Few Hours Remaining for Rebuttal Phase**
>
> Dear Reviewer,
>
> I hope this message finds you well. I am writing to kindly remind you that the rebuttal phase for our paper is due in a few hours.
>
> We fully understand the commitments and busy schedule you may have, and we sincerely appreciate your time and effort put into this process.
>
> We are eager to learn if we have adequately addressed your concerns and whether you have any further questions. Your feedback is instrumental in enhancing the quality of our paper.
>
> Thank you once again for your contribution and understanding.

---

### Official Review · Reviewer_AdWg · 2023-11-01

**Soundness:** 3 good
**Presentation:** 3 good
**Contribution:** 3 good
**Rating:** 8
**Confidence:** 4

**Summary:**

The paper studies approximation properties of residual networks under usual assumptions. Given a function space, input dimension, and a precision requirement, upper and lower bounds on the number of parameters of a ResNet architecture are studied. These bounds are derived to ensure the function family parameterized a ResNet architecture is dense in the function space under the supremum norm. By extending a result by Yarotsky (2017), a lower bound for the size of a ReLU network which approximates a multivariate product function is first derived. Since there is always a ReLU network which represents a ResNet, a lower bound in ReLU networks also serves as a lower bound for ResNets. As the family of multivariate product function is a subset of polynomials, the lower bound derived is also applicable to polynomials. Then, upper bounds are derived for monomials, which is smaller by a factor of input dimension d compared to the upper bound on the family of ReLU networks. Next, the bound is extended to polynomials and the Sobolev space. The bound is also smaller by a factor of d compared to the one on ReLU networks. Finally, it was shown that any continuous piecewise linear function can be approximated by a ResNet with width $(d+1)$ when the depth $L$ is large enough. Then, the authors argue that ResNets can approximate any continuous function. A relation between ResNets and dynamic systems is also studied. By limiting the regularity of the outer function of the Kolmogorov representation of a continuous function, the authors show that any function in this special family can be approximated by ResNets in the uniform norm.
In addition to the above theoretical results, the authors also demonstrate the approximation ability of ResNets by changing their size in computer simulation where mean squared error is used to measure the approximation capability.

**Strengths:**

- Originality: Most existing approximation bounds are derived for networks without skip connections. Hence, the proposed upper and lower bounds are novel in the sense that they are tailored to the ResNet architecture.

- Quality and clarity: This paper gives a comprehensive presentation on the approximation properties of ResNet. The background knowledge is well-organized, and the theoretical results are presented in a flow that is easy to follow and understand.

- Significance: ResNet is an important architecture and understanding its approximation limitations and abilities is crucial.

**Weaknesses:**

1. The upper bounds for monomial and polynomials (Theorem 3 and 4) in this paper are not surprising in the sense that any layer in a ReLU network can be configured to implement a d-dimensional identity function if the width is sufficiently large. Since a $d$-dimensional identity function can be represented by a layer of $2d$ ReLU units, a smaller upper bound (for approximating monomials) by a factor of $d$ is expected. The upper bound given in Theorem 5 is the same as the bound given in Theorem 1 by Yarotsky (2017). Although this is for ResNet, it is not surprising using the above argument. The large constant d is also hidden in the big O notation. This piece of result can be more interesting if the authors can show the dependency on d explicitly and demonstrate that it is also some factors of reduction similar to case of monomials.

2. For the lower bound (Theorem 2), it is not very useful given the product function is limited. The authors give a comprehensive discussion but the result extending this bound to the space of continuous functions is missing. It is good to discuss existing results, but a comparison is expected.

3. In Theorem 6, the upper bound for approximating continuous functions grows with the input dimension given that the width is bounded by $(d+1)$, provided the depth is sufficiently large. Such dependency on the input dimension is usually not friendly given that many neural networks in applications directly work on raw data. A dimension-independent bound (reference below) can be derived when the number of pieces in the continuous piecewise linear function is known.

>Chen, Kuan-Lin, Harinath Garudadri, and Bhaskar D. Rao. "Improved bounds on neural complexity for representing piecewise linear functions." Advances in Neural Information Processing Systems 35 (2022): 7167-7180.

4. On the other hand, the depth is not explicitly given in the theorem, which even reduces the significance of the statement. It would be clearer if the authors could explain this missing part in the discussion following Theorem 6.

5. The function space considered in Theorem 8 is very limited. Such a space is even smaller than the space used by Theorem 4.1 in the following paper.

>Montanelli, Hadrien, and Haizhao Yang. "Error bounds for deep ReLU networks using the Kolmogorov–Arnold superposition theorem." Neural Networks 129 (2020): 1-6.

6. In the above paper, the space of functions is defined based on the refinement levels of the inner functions which can be used to bound the Lipschitz constant of the outer functions. However, Theorem 8 directly limits the Lipschitz constant of the outer function. This assumption is unrealistic, and it avoids the main difficulty in approximating the outer function.

6. Since the approximation quality is measured by the uniform norm, it seems not reasonable to measure the mean squared error. Measuring the maximum error could be more convincing.



### post-rebuttal

Many of my concerns have been addressed, and in light of these improvements, I have revised my rating from 5 to 8.

**Questions:**

1. In the second paragraph after Proposition 1, can one of the $\alpha_i$ be 0?

2. It would be clearer if the authors can state that this paper only discusses bounds under the uniform norm. Would it be possible to derive tighter bounds under $L^2$ norm for ResNet?

3. The variable d is missing in the last sentence of Theorem 3.

4. Theorem 5. There is a ResNet R that can … A word is missing.

5. The paragraph following Theorem 5, please clarity that the bound is nearly tight up to a log factor of epsilon.

6. The statement after “Thus” of Theorem 6 is incomplete.

7. The last paragraph on page 7. Can you provide some references to justify why ResNet has superior optimization performance? Do you mean by the ability to improve representation? Are there any guarantees?

8. The authors claim that experiment results show the exceptional approximation capability of ResNet for learning complex functions. Can you evaluate the trained ResNets using the maximum errors?

9. It would be clearer if the authors can emphasize in the conclusion that the class of smooth functions is a limited class.

---

> ### Author Response · Authors · 2023-11-18
> **Reply from authors**
>
> Dear reviewer,
>
> Thank you for the time and effort in giving our paper a thorough review. We are very happy to receive your comments and suggestions. For your concerns and questions, we address them one by one as follows.
>
> ## Response to Concerns and Comments-Part 1
>
> **Comment 1.1**: *The upper bounds for monomial and polynomials (Theorem 3 and 4) in this paper are not surprising in the sense that any layer in a ReLU network can be configured to implement a d-dimensional identity function if the width is sufficiently large. Since a d-dimensional identity function can be represented by a layer of 2d ReLU units, a smaller upper bound (for approximating monomials) by a factor of d is expected.*
>
> Response: There may be a confusion/misunderstanding, and we would like to clarify it in the following.
>
> It is true that representing a $d$-dimensional identity function via a layer of $2d$-ReLU units in ResNet implies an *additive* reduction of $d$ tunable weights compared to an FNN when approximating the same function. However, this does not inherently suggest a *multiplicative* reduction by a factor of $d$. For instance, suppose ResNet requires $d$ tunable weights per residual block to approximate a function with the same accuracy level as a ReLU network necessitating $3d$ tunable weights, the reduction factor is indeed $3$, not $d$.
>
> In our study, however, this additive reduction of $d$ tunable weights, as seen in our b-ResNet model, does translate into a multiplicative reduction by a factor of $d$. We establish this result by proving that a b-ResNet with a constant number of tunable weights per residual block can approximate functions with the same accuracy as a ReLU FNN requiring $O(d)$ tunable weights. Our proof is constructive, and we believe this result adds valuable and non-trivial insights to our understanding of these networks. These discussions have been added in the revised version (after Theorem 3, 'Root of reduction' bullet point, **Sec. 4.1 Page 6-7**).
>
> **Comment1.2**: *The upper bound given in Theorem 5 is the same as the bound given in Theorem 1 by Yarotsky (2017). The large constant d is also hidden in the big O notation. This piece of result can be more interesting if the authors can show the dependency on d explicitly and demonstrate that it is also some factors of reduction similar to case of monomials.*
>
> Response: We appreciate your attention to detail. To clarify, the upper bound presented in Theorem 5 does indeed have the same expression in terms of $\epsilon$ as that in Yarosky (2017). However, it is crucial to note that the expression in terms of '$d$' is a factor of '$d$' smaller than Yarosky's. This is due to our approach of approximating smooth functions using polynomials.
>
> As previously discussed, this reduction cannot be directly inferred from the argument concerning additive reduction. We consider the result presented in Theorem 5 non-trivial and it contributes substantially to the literature on ResNet performance analysis. Moreover, it fill in the research gap for the approximation capability of ResNet in Sobolev space.
>
> Even though we can not explicitly say if there is some factors of reduction in Theorem 5, we have given bounds for the hidden constant (following theorem 5, **Sec.4.2 Page 7**) in the revised version of our paper in order to enhance clarity and ease of comprehension.

---

> ### Author Response · Authors · 2023-11-18
>
> ## Response to Concerns and Comments-Part 2
>
> **Comment 2**: *For the lower bound (Theorem 2), it is not very useful given the product function is limited. The authors give a comprehensive discussion but the result extending this bound to the space of continuous functions is missing. It is good to discuss existing results, but a comparison is expected.*
>
> Response: Thanks for your attention to the details. We first note that the results in Theorem 2 establish a tight lower bound for NN to approximate product functions, and hence for polynomial functions. Such a tight lower bound is crucial for showing the upper bound of ResNet's approximation of polynomial functions in Theorem 4 is order optimal (in `ε`) and such a tight lower bound for NN approximation of polynomial functions is not available in the literature and directly using other lower bounds developed for general functions, e.g., those for continuous functions, does not serve our purpose.
>
> To this end, we would also like to clarify a confusion probably due to the way we organize Theorem 2 and the discussions below it. Our intended logic flow in Sec. 3 (Lower bounds on ResNet's approximation capability), after showing the ResNet can be implemented by an FNN (with extra tunable weights), is to (i) state that all lower bounds for FNN's approximation capability for different function classes can then be easily adapted to ResNets, and (ii) for a special class of functions, polynomials, we derive a tight lower bound. Such a lower bound matches the upper bound in Theorem 4, which will be mentioned following theorem 4.
>
> Last, we would like to note that there seems little room for extending our tight lower bound for polynomial functions to more general functions, e.g., continuous functions, as the existing lower bounds for continuous function is already proven to be order optimal (ref here). Nonetheless, this may be some room in improving the constants in the lower bounds, and we have included it as an interesting future direction in the conclusion section of the revised manuscript.
>
> We have revised the **Sec. 3** in the revised manuscript to make the above points clear and avoid potential future confusion; see **page 5-6**. Many thanks for your helpful comments!
>
> **Comment 3**: *In Theorem 6, the upper bound for approximating continuous functions grows with the input dimension given that the width is bounded by `d+1`, provided the depth is sufficiently large. Such dependency on the input dimension is usually not friendly given that many neural networks in applications directly work on raw data. A dimension-independent bound (reference below) can be derived when the number of pieces in the continuous piecewise linear function is known.*
>
> *Chen, Kuan-Lin, Harinath Garudadri, and Bhaskar D. Rao. "Improved bounds on neural complexity for representing piecewise linear functions." Advances in Neural Information Processing Systems 35 (2022): 7167-7180.*
>
> Response: Thank you for your thoughtful comment and the suggested reference. We've thoroughly reviewed the paper you mentioned and conjecture ResNet will have a similar result that a dimension-independent bound could be derived when the number of pieces in the continuous piecewise linear function is known. This indeed offers an interesting perspective on our results. In response to your feedback, we have included this limitation and related discussion following Theorem 6 (**Page 8**) in the updated version of our paper. We believe this will enrich our analysis and provide a more comprehensive view to our readers.
>
>
> **Comment4**: *On the other hand, the depth is not explicitly given in the theorem, which even reduces the significance of the statement. It would be clearer if the authors could explain this missing part in the discussion following Theorem 6.*
>
> Response: Thanks for your suggestion. The depth L=$O(Md)$ where $M$ is an $f$-dependent parameter. Please refer to Thm 6. and the clarification and discussion have been added in the revised manuscript following **Theorem 6, Page 8.**

---

> ### Author Response · Authors · 2023-11-18
>
> ## Response to Concerns and Comments - Part 3
>
> **Comment 5 and 6**: *The function space considered in Theorem 8 is very limited. Such a space is even smaller than the space used by Theorem 4.1 in the following paper. In the above paper, the space of functions is defined based on the refinement levels of the inner functions which can be used to bound the Lipschitz constant of the outer functions. However, Theorem 8 directly limits the Lipschitz constant of the outer function. This assumption is unrealistic, and it avoids the main difficulty in approximating the outer function.*
>
> A: We acknowledge this function space in Theorem 8 is limited. The motivation behind our approach in Theorem 8 was to explore conditions under which we could overcome the curse of dimensionality by using ResNet to approximate the inner and outer functions separately. In this context, limiting the Lipschitz constant of the ResNet function appeared to be a viable strategy, which led us to directly constrain the Lipschitz constant of the outer function.
>
> We understand that this assumption may seem unrealistic and bypasses the central challenge in approximating the outer function. Your feedback gives us a valuable perspective on this, and we have provided further discussion in the revised version of our paper. Please refer to **Paragraph 2 following Theorem 7, Page 8-9.**
>
> **Comment 7**: *Since the approximation quality is measured by the uniform norm, it seems not reasonable to measure the mean squared error. Measuring the maximum error could be more convincing.*
>
> A: Your feedback on our choice of error measurement is valued. The reason we initially opted for the mean squared error is its widespread acceptance and preferred use in practice. However, we understand the limitations you have pointed out and agree that measuring the maximum error could provide a more convincing evaluation of our approach. We have supplemented our experimental results with measurements using the max error in the revised version of our manuscript (see **Sec. 6, Page 9** and Appendix H).

---

> ### Author Response · Authors · 2023-11-18
>
> ## Response to the questions-Part 1
> We appreciate your time and effort in providing us with these insightful questions and suggestions. We will address them systematically. Your recommendations concerning the writing style and clarity (Questions (2)(4)(5)(6)(9)) are well taken. All necessary clarifications and modifications in line with your suggestions are incorporated into our revised version.
>
> As for the other questions, we will address each one individually as follows:
>
> **Q1**: *In the second paragraph after Proposition 1, can one of the $\alpha_i$ be 0?*
>
> Response: Yes, any $\alpha_i$ can be $0$.
>
> **Q2**: *It would be clearer if the authors can state that this paper only discusses bounds under the uniform norm. Would it be possible to derive tighter bounds under $L_2$ norm for ResNet?*
>
> Response: Thanks for the suggestion, we have added a clarification in **section 2.2**. Regarding your second question, we have the following explanation. Considering that the target function is continuous, the uniform norm is equivalent to the $L_\infty$ norm, thus encompassing the $L_2$ norm. This means that if $|f-g|_ {L_ \infty}<\epsilon$, then $|f-g|_ {L_2}<c\epsilon$ also holds, where $c$ is some constant. Typically, using the $L _2$ norm wouldn't alter the upper bounds in terms of $\varepsilon$. However, it could potentially make the hidden constant ($C(d)$) in the upper bounds ($ \mathcal{O}(C(d)g(\varepsilon,d)) $) tighter for more complex function spaces, such as smooth or continuous functions. The results of the following reference will be useful for this question.
>
> Lu, J., Shen, Z., Yang, H., \& Zhang, S. (2021). Deep network approximation for smooth functions. SIAM Journal on Mathematical Analysis, 53(5), 5465-5506.
>
> **Q3**: *The variable $d$ is missing in the last sentence of Theorem 3.*
>
> Response: The variable $d$ is indeed not missing. For a $d$-dimension monomial with degree $p$, if $p< d$, then it can be viewed as a monomial of dimension $p$. So in general case, we always consider $p\geq d$. Consequently, the upper bound in Theorem 3 is not directly dependent on $d$. It instead primarily depends on the degree $p$, which, you could argue, inherently relates to the dimension $d$ or the influence of $d$ is embedded in the degree $p$. We have included some comments following Theorem 3 (**Page 6**) in the revised manuscript to clarify this point and avoid any potential confusion.
>
> **Q4**: *Theorem 5. There is a ResNet R that can … A word is missing.*
>
> Response: Thanks for the notice. We have checked typos and grammar errors in our paper and have modified them in the revised version.
>
> **Q5**: *The paragraph following Theorem 5, please clarity that the bound is nearly tight up to a log factor of epsilon.**
>
> Response: We appreciate your suggestion for clarification. The lower bound of the complexity (number of weights) required for an FNN to approximate a smooth function is $\Theta(\varepsilon^{-d/r})$ as per Theorem 3 in Yarotsky (2017). From Proposition 1 in our paper, we know that this lower bound can also be applied to ResNets. Our derived upper bound is $\mathcal{O}(\varepsilon^{-d/r}\log 1/\varepsilon)$ for ResNets. Considering these two bounds, we can say that our upper bound is nearly tight up to a logarithmic factor of $\varepsilon$. We have incorporated this clarification in the paragraph following Theorem 5 (**Page 7**) in the revised version of our manuscript.
>
> **Q6**: *The statement after “Thus” of Theorem 6 is incomplete.*
>
> Response: We acknowledge this word is not incomplete. Here 'Thus' means that we use piece-wise spline function to approximate continuous function. The conclusion after 'Thus' is the second point of Thm. 6 and the proof can be found in the Appendix. Moreover, the clarification has been made in the revised version.
>
> **Q7**: *The last paragraph on page 7. Can you provide some references to justify why ResNet has superior optimization performance? Do you mean by the ability to improve representation? Are there any guarantees?*
>
> Response: There is some clarification for the question. For the optimization performance, we refer to the high training efficiency of ResNet. Training efficiency primarily stems from the architecture's ability to mitigate issues related to gradient descent, such as vanishing or exploding gradients, which are common challenges in deep learning. The skip connections effectively alleviate the aforementioned issues and contribute to improved training performance. These benefits of ResNet architecture were discussed in the original ResNet paper by He et al. (2016). Note that this paragraph about these discussions has been placed in **Appendix E.2** due to the limitations of paper space.
>
> Kaiming He, Xiangyu Zhang, Shaoqing Ren, and Jian Sun. Deep residual learning for image recognition. In Proceedings of the IEEE conference on computer vision and pattern recognition, pp. 770–778, 2016.

---

> ### Author Response · Authors · 2023-11-18
>
> ## Response to the questions - part 2
> **Q8**: *The authors claim that experiment results show the exceptional approximation capability of ResNet for learning complex functions. Can you evaluate the trained ResNets using the maximum errors?*
>
> Response: Sure. We have supplemented our experimental results with measurements using the max error in the revised version (See **Sec. 6 Page 9** and Appendix H) of our manuscript.
>
> **Q9**: *It would be clearer if the authors can emphasize in the conclusion that the class of smooth functions is a limited class.*
>
> Response: Your suggestion is well taken. The smooth function space, specifically the Sobolev space as discussed in our paper, is a highly important class of functions and it includes the Lipschitz continuous functions. Its relevance is particularly pronounced in mathematics, where, for instance, weak solutions to several essential partial differential equations exist within suitable Sobolev spaces. In practical applications, Lipschitz continuous functions are quite common, making it crucial to understand the approximation capability of ResNet for this class of smooth functions. Due to the limit of paper space, the conclusion part is removed. Thus, we commented on this point **following Thm.5, Page 7 (last sentence in the paragraph)**.

---

> > ### Comment · Reviewer_AdWg · 2023-11-21
> >
> > I would like to thank the authors for their detailed and thoughtful response. Most of my concerns are addressed. After reviewing the revised paper, I have decided to increase my overall rating from 5 to 8 as well as the scores for the quality and contribution.
> >
> > Additional comments to the authors' response:
> >
> > 1.	I was actually more interested in how one can theoretically prove that the optimization landscape of ResNet is better than the corresponding FNN in the sense of avoiding the so-called degradation problem. However, given the claim has been moved to Appendix E.2 with proper justification, this concern has been addressed.
> > 2.	Since the proof is constructive, does it make more sense to directly construct the approximation network rather than applying SGD? Additionally, what would be the computational complexity of the constructive algorithm, and is it expected to be more efficient than SGD? While these questions are intriguing, they may fall outside the scope of the current paper.
> > 3.	Could you kindly elucidate the reason behind the superior performance of the ReLU network as depicted in Figure 2 (d=100) when compared to ResNets?
> > 4.	I think it is good to show plots with MAX loss even though they have large errors. Did you train these models using MSE and then evaluate on both MSE and MAX? Are you able to train them using the MAX loss?

---

> ### Author Response · Authors · 2023-11-22
>
> ## Response to the additional questions 1/2
>
> Dear reviewer,
>
> We are very grateful for your time and effort in reviewing our revised paper. Your positive reception and the increase in your rating are encouraging, and we're delighted to know that we have addressed most of your concerns. Regarding your additional comments and questions, we address them one by one in the following.
>
> **Comment 1**: *I was actually more interested in how one can theoretically prove that the optimization landscape of ResNet is better than the corresponding FNN in the sense of avoiding the so-called degradation problem. However, given the claim has been moved to Appendix E.2 with proper justification, this concern has been addressed.*
>
> Response to Comment 1: We're glad to hear that the proper justification in Appendix E.2 has addressed your concern. Please do not hesitate to reach out if you have further questions or need additional clarification on this or any other aspect of our work. Thanks you for your insightful and instructive comments again.
>
>
> **Q2**: *Since the proof is constructive, does it make more sense to directly construct the approximation network rather than applying SGD? Additionally, what would be the computational complexity of the constructive algorithm, and is it expected to be more efficient than SGD? While these questions are intriguing, they may fall outside the scope of the current paper.*
>
> Response to Q2:
>
> Thank you for this interesting question. Here's our perspective on these points:
>
> Just as in all constructive proofs stipulated in the literature for establishing the universal approximation capability of DNN/CNN/RNN, our method of constructing a ResNet to precisely approximate a function also necessitates the explicit form of this function. However, in practical applications, we typically do not possess the explicit form of the function that we aim to learn through Neural Network (NN) training -- it remains unknown and is what we seek to discover. If we already knew the explicit form of the function, there would be no requirement to train an NN to learn it approximately.
>
> Our results on the universal approximation capability of ResNet, much like those for DNN/CNN/RNN in the literature, serve to assure practitioners that a ResNet with certain parameters exists, which can accurately approximate the target function. In practice, we strive to discover these parameters by training a ResNet with an adequate number of neurons using Stochastic Gradient Descent (SGD) or other optimization algorithms.
>
> Without such a universal approximation capability, practitioners might remain uncertain whether a certain function can be approximated by a DNN/ResNet with a specific number of tunable parameters, even if the learning process is executed flawlessly. Thus, from this perspective, the establishment of the universal approximation capability for DNN/ResNet carries significant value.

---

> ### Author Response · Authors · 2023-11-22
>
> ## Response to additional questions 2/2
>
>
> **Q3**: *Could you kindly elucidate the reason behind the superior performance of the ReLU network as depicted in Figure 2 (d=100) when compared to ResNets?*
>
> Response to Q3: We appreciate your astute observation. Our understanding is that for the case of $d=100$, both the ReLU and ResNet appear proficient in learning the target function well, as evidenced by their loss performance. However, as we escalate the dimension from 100 to 300, the ReLU network appears to struggle with learning the target function, as suggested by the non-converging loss. In contrast, ResNet maintains a small training loss upon convergence, which signifies that it is still capable of effectively learning the target function.
>
> **Q4**: *I think it is good to show plots with MAX loss even though they have large errors. Did you train these models using MSE and then evaluate on both MSE and MAX? Are you able to train them using the MAX loss?*
>
> Response to Q4: Thanks for your insightful question. In response, we have conducted new experiments related to training models using MAX loss, and then evaluating them on both MSE and MAX metrics. Please refer to **Appendix H on page 32** for the associated figures.
>
> In summary, we found that models trained with MSE loss exhibit superior testing performance, as shown in both lower MSE and MAX testing losses, compared to those trained with MAX loss.
> This observation indeed aligns with our understanding. Training with MAX loss focuses on a single data item during each iteration of its loss function would result in a "local" learning process that may not fully capture the function over its support set.
> Conversely, training with MSE loss minimizes the loss across a batch of data items at each iteration, enabling more global learning of the function over its domain and subsequently reducing the approximation error.
>
> This phenomenon also aligns with prevalent empirical experiences suggesting that training with average loss tends to be more stable than training with worst-case loss. As a result, MSE loss is often favored over MAX loss in practical engineering applications, due to its greater stability and strong performance.
>
> Thanks again for your time and effort!

---

> > ### Comment · Reviewer_AdWg · 2023-11-23
> >
> > I wish to extend my sincere gratitude to the authors for taking the time to address my additional inquiries. Having no further questions, I am confident in asserting that this paper stands as a noteworthy contribution to the field.

---

> > > ### Author Response · Authors · 2023-11-23
> > > **Thanks for Reviewer Feedback and Appreciation for Constructive Review Process**
> > >
> > > Dear Reviewer,
> > >
> > > Thank you for your kind words and positive feedback. We deeply appreciate the time and effort you've invested in reviewing our paper. Your constructive feedback and insightful comments have greatly helped improve our work.
> > >
> > > Once again, thank you for your support and for the valuable role you've played in this process.
> > >
> > > Best regards

---

### Official Review · Reviewer_gSxo · 2023-11-01

**Soundness:** 3 good
**Presentation:** 3 good
**Contribution:** 3 good
**Rating:** 6
**Confidence:** 4

**Summary:**

The paper aims to provide a theoretical understanding of the expressive power of Resnets. The authors study this question w.r.t. function approximation. They first start with d-dimensional monomials of degree p, then they study smooth functions that are differentiable up to degree r, and they also provide a class of lower bounds. Their main results are to derive bounds on the number of tunable weights needed or sufficient for the function approximation. The authors' results showcase the benefits of ResNets compared to ReLU nets in terms of necessary tunable weights.

The main idea behind the results is to show a connection between ResNets and their feedforward counterparts. The authors show that ResNets can be viewed as a sparse FNN. The authos also provide specific constructions that show how to approximate classes of functions mentioned before with a bounded (and they provide the bounds) number of tunable weights.

**Strengths:**

+well-motivated question about the theoretical properties of ResNets

+clean results in that the bounds are interesting

**Weaknesses:**

Overall i like the paper however, I find the most important weakness to be w.r.t. novelty and technical innovation.

-the main concern is that many of the ideas/constructions upon which the paper relies to prove their results have been known in prior works. This is the case with the proposed lower bound for example. The most interesting observation I would say is the implementation of quadratics "x^2" and product "xy" using ResNets, but I don't find this contribution enough for ICLR.

**Questions:**

Q: A related interesting question is about the depth-width tradeoffs for ResNets. Perhaps this is something that follows from your work on just the tunable weights, but as far as I can tell it's not obvious. Given that there are many results for FNNs and their depth/width tradeoffs,  I believe it is important to highlight this too.

---

> ### Author Response · Authors · 2023-11-18
> **Reply from authors**
>
> Dear Reviewer,
>
> Thanks for your time and effort in reviewing our paper. We are pleased to know that you find our question well-motivated and our results interesting. We value your constructive feedback and there are some clarifications and feedback for your concerns and questions.
>
> **1) Concerns about the novelty and technical innovation.**
>
> Response: Thanks for the note and we clarify in the following. Our paper builds upon existing techniques with new ideas to establish the universal approximation capability of ResNet FNN, addressing an important research gap in the literature and add to the theoretical justification of ResNet's excellent practical performance. While some techniques used in our paper are known in prior works, most proof ideas for existing approximation bounds are derived for networks without exploiting the skip connections unique to ResNet FNN. Our work extends these techniques to the case with skip connections, which involve some non-trivial development.
>
> Particularly, in our construction, we leverage the identical mappings given by the skip connections in the construction of ResNets to approximate multivariate functions, which is the key to show that b-ResNet (ResNet with constant width) has powerful expressivity. This is one of our novel technical innovations in this paper.
>
> Furthermore, We also prove that ResNet with single-neuron can approximate any continuous piece-wise linear function. The result is a non-trivial extension of the result in Lin & Jegelka (2018), which shows ResNet with single-neuron can approximate step functions. We also show how ResNet can be incorporated into a novel NN architecture suggested by the KST theorem to efficiently approximate a class of smooth functions.
>
> All the above results are the first in the literature, addressing important research gaps and adding to the literature on ResNet's performance analysis.
>
> **2) Questions about the depth-width tradeoffs for ResNets.**
>
> Response: Thanks for this insightful question! We are well aware of the depth-width trade-off in FNN's universal approximation capability as in the classical papers [1,2,3].
>
> In this paper, we focus on a b-ResNet (ResNets with constant width) and show even this simple ResNet can achieve strong universal approximation capability. We briefly discuss the performance of deep ResNet and shallow ResNet following theorem 3, (at the 'Deep vs. shallow' bullet point, **Page 6-7**), in that deep ResNet can achieve better approximation error than the shallow one on the approximation of monomials/polynomials but maybe the function space is limited. We believe that a more thorough investigation on ResNet's depth-width tradeoffs is both practically important and theoretically interesting, and we have noted it as a future direction in the revised version of the paper.
>
> [1] Shiyu Liang and Rayadurgam Srikant. Why deep neural networks for function approximation? arXiv
> preprint arXiv:1610.04161, 2016.
>
> [2] Dmitry Yarotsky. Error bounds for approximations with deep relu networks. Neural Networks, 94:
> 103–114, 2017
>
> [3] Safran, Itay, and Ohad Shamir. Depth-width tradeoffs in approximating natural functions with neural networks. International conference on machine learning. PMLR, 2017.

---

> ### Author Response · Authors · 2023-11-22
> **Gentle reminder**
>
> Dear reviewer,
>
> We hope this message finds you well. We are writing to kindly remind you that the rebuttal stage deadline is due in a day. We fully understand the  busy schedule you may have, and we sincerely appreciate your time and effort put into this process. We are eager to learn if we have adequately addressed your concerns and whether you have any further questions. Your feedback is instrumental in enhancing the quality of our paper. Thanks for your time and effort again!

---

> ### Author Response · Authors · 2023-11-23
> **Gentle Reminder: Few Hours Remaining for Rebuttal Phase**
>
> Dear Reviewer,
>
> I hope this message finds you well. I am writing to kindly remind you that the rebuttal phase for our paper is due in a few hours.
>
> We fully understand the commitments and busy schedule you may have, and we sincerely appreciate your time and effort put into this process.
>
> We are eager to learn if we have adequately addressed your concerns and whether you have any further questions. Your feedback is instrumental in enhancing the quality of our paper.
>
> Thank you once again for your contribution and understanding.

---

### Author Response · Authors · 2023-11-18
**Revised Version Uploaded**

We have uploaded the revised version of our paper. The main differences from the previous version are as follows:

1. Some typos, and grammatical errors have been corrected to improve readability.
2. Additional clarifications and discussions have been added (marked in blue) in response to the suggestions and comments from the reviewers. More details are in the Reply to each reviewer.
3. Due to the limit of paper space, Some discussions (including Sec. 4.4, the detailed information of related work) have been relocated to the Appendix. Moreover, the Conclusion part (Sec. 7) is removed.
4. For better readability, the attached PDF contains both main text and Appendix.

Minor comments: Some writing techniques have been adjusted in Sec. 2 to improve readability.

We note that all the revised and newly-included contents are marked in color **blue**, for easy identification.

We express our thanks to all the reviewers for their instructive comments and valuable feedback.

---

### Author Response · Authors · 2023-11-20
**Gentle reminder**

Dear Reviewers,

We hope this message finds you well. We wanted to gently remind you that we are approaching the deadline for our rebuttal process, which is in two days. We have made significant revisions to our paper based on your valuable feedback and are eager to hear your thoughts on these changes.

Your insights have been instrumental in improving the quality of our work, and we would greatly appreciate your further comments to ensure we have adequately addressed all your concerns.

We understand that you are busy, and we genuinely appreciate the time and effort you are investing in this review process.

Thank you once again for your time and effort!

---

### Meta-Review · Area_Chair_1sbi · 2023-12-12

**Metareview:**

The manuscript studies the universal approximation of resnet, while this problem is well studied, the paper contributes to some different aspects of the problem. Overall, the AE feels that such study of approximation property does not really shed much light on the practical usefulness of the architecture, and overall there has been quite extensive study of approximation power already, as pointed out by some expert reviewers, the contributions compared to some previous literature is marginal.

**Justification For Why Not Higher Score:**

Approximation results do not really explain the powerfulness of neural networks in practice, and compared with previous studies, the contribution is incremental.

**Justification For Why Not Lower Score:**

N/A

---

### Decision · Program_Chairs · 2024-01-16

Reject